# InfiGUI-R1:
# Advancing Multimodal GUI Agents from Reactive Actors to Deliberative Reasoners

## Abstract

Multimodal Large Language Models (MLLMs) have shown significant promise in powering Graphical User Interface (GUI) agents to automate complex digital tasks. However, the prevailing monolithic training paradigms often create a structural mismatch with the hierarchical nature of capabilities required for robust performance. Specifically, the efficacy of methods like Reinforcement Learning (RL) is critically predicated on the agent possessing a high-quality behavioral prior of key reasoning skills, such as spatial reasoning and goal decomposition, which are often absent. To resolve this impasse, we propose **Actor2Reasoner**, a novel two-stage hierarchical training paradigm grounded in the principle of *Endow First, Internalize Later*. The first stage, **Cognitive Endowment**, employs targeted supervised fine-tuning to instill these crucial thinking patterns, forging a *Capable Actor*. Subsequently, the second stage, **Policy Internalization**, utilizes RL to evolve this actor into a *Deliberative Reasoner* by internalizing the endowed abilities into a robust, context-aware decision-making policy. We instantiate our paradigm in **InfiGUI-R1**, an agent that achieves state-of-the-art performance on challenging benchmarks, including AndroidControl. Our work demonstrates that decoupling the endowment of foundational abilities from the internalization of policy provides a more effective and principled path toward developing sophisticated and resilient GUI agents.

## 1 Introduction

The widespread adoption of Graphical User Interfaces (GUIs) has made digital devices universally accessible, yet mastering the vast landscape of applications remains a significant cognitive load for users. GUI agents, particularly those powered by recent advances in Multimodal Large Language Models (MLLMs) (Liang et al., 2024; Peng et al., 2023; Awadalla et al., 2023; Li et al., 2024a; Wang et al., 2024), offer a compelling vision to alleviate this burden by automating complex digital workflows (Bonatti et al., 2024; Rawles et al., 2024). While MLLMs can leverage their pretrained knowledge to recognize visual elements, realizing true autonomy requires a deeper layer of GUI-centric reasoning. For instance, successfully booking a ride-sharing service requires not just identifying the 'destination' field, but also employing key thinking patterns: using spatial reasoning to locate it accurately on a cluttered screen, using goal decomposition to understand that 'entering the destination' is the immediate necessary step, and applying reflection to verify the action's outcome. This reveals an inherent hierarchy of capabilities: a foundation of key GUI reasoning patterns supports a superstructure of strategic, task-oriented policy.

Many current training paradigms, however, employ a monolithic approach that fails to respect this capability hierarchy, creating a fundamental mismatch that limits agent performance. On one hand, Supervised Fine-Tuning (SFT) on expert trajectories (Lin et al., 2024; Cheng et al., 2024) is effective for teaching agents basic operational skills, but struggles to instill the adaptive, strategic reasoning required for novel or complex scenarios. On the other hand, reasoning-enhanced Reinforcement Learning (RL), such as methods leveraging verifiable rewards (RLVR), has emerged as a promising direction for developing effective reasoning ability. Yet, the efficacy of RL is predicated on a critical assumption: that the underlying model already possesses, and can produce with non-trivial probability, thoughts and actions corresponding to key reasoning capabilities such as spatial under-

standing, sub-goal decomposition, and reflection. If an MLLM lacks these foundational abilities in its behavioral prior, RL's exploration process may struggle to discover them from scratch within a practically infinite action space, especially for MLLMs with a smaller parameter scale, leading to an intractable learning problem. Furthermore, another line of work involving large-scale online RL, while potent, faces significant hurdles including prohibitive costs and the absence of authoritative open-source frameworks, limiting its widespread adoption.

To resolve this structural impasse, we argue for a paradigm shift away from monolithic training. We propose **Actor2Reasoner**, a novel, two-stage hierarchical training paradigm built on the principle of *Endow First, Internalize Later*. This paradigm replaces the single, overloaded learning objective with a deliberate, architectural progression. The first stage, **Cognitive Endowment**, uses targeted supervised fine-tuning on a curated dataset to instill a robust behavioral prior. Its explicit goal is to equip the agent with the key cognitive capabilities essential for GUI navigation, such as spatial reasoning for element localization and task decomposition for processing instructions. This stage forges a *Capable Actor*. Only with this endowed foundation does the second stage, **Policy Internalization**, begin. Here, the Capable Actor is evolved into a *Deliberative Reasoner*. Using reinforcement learning with specific mechanisms like sub-goal guidance and error recovery scenarios, the agent learns to internalize a robust, context-aware decision-making policy. By decoupling the endowment of foundational abilities from the internalization of contextual policy, our paradigm ensures that policy learning becomes a focused process of refining a competent behavioral prior, rather than an inefficient, from-scratch search for basic skills.

We instantiate this paradigm in InfiGUI-R1, an agent whose performance validates our architectural principles. Our contributions are:

- We identify a structural mismatch in many current GUI agent training approaches, where monolithic paradigms conflict with the hierarchical nature of required capabilities, and we highlight the dependency of effective RL on an agent's pre-existing foundational reasoning skills.
- We propose Actor2Reasoner, a new hierarchical training paradigm that resolves this mismatch by first endowing an agent with key cognitive abilities (Cognitive Endowment) before refining its decision-making with RL (Policy Internalization) (§3).
- Through our agent, InfiGUI-R1, we provide a concrete instantiation of our paradigm. Extensive experiments not only set a new state-of-the-art on challenging benchmarks but also, via rigorous ablation studies, empirically validate the architectural necessity and synergistic benefits of our two-stage design (§4).

## 2 RELATED WORK

### 2.1 MULTIMODAL LLMS

Large Language Models (LLMs) (Floridi & Chiriatti, 2020; Touvron et al., 2023; Bai et al., 2023a; Xiao et al., 2021) have significantly enhanced the capabilities of AI systems in tackling a wide range of tasks (Hu et al., 2024b; Li et al., 2024b), thanks to their exceptional ability to process complex semantic and contextual information. The remarkable power of LLMs has also inspired exploration into their potential for processing multimodal data, such as images. Typically, the architecture of Multimodal Large Language Models (MLLMs) consists of three main components: a pre-trained large language model, a trained modality encoder, and a modality interface that connects the LLM with the encoded modality features. Various vision encoders, such as ViT (Dosovitskiy et al., 2021), CLIP (Radford et al., 2021), and ConvNeXt (Liu et al., 2022), extract visual features, which are integrated using techniques like adapter networks (Liu et al., 2023), cross-attention layers (Alayrac et al., 2022), and visual expert modules (Wang et al., 2023). These methods have facilitated the development of high-performing MLLMs, such as Qwen-VL (Bai et al., 2023b), GPT-4 Vision (OpenAI, 2023), BLIP-2 (Li et al., 2023) and InfiMM (Liu et al., 2024), thus opening new avenues for LLMs in processing GUI tasks.

### 2.2 MLLM-BASED GUI AGENTS

Agents are AI systems that perceive their environments, make decisions, and take actions to complete specific tasks. The emergence of LLMs with human-level reasoning ability has significantly advanced the development of agents. For GUI tasks, earlier systems relied on LLMs to read and

interpret structured representations such as HTML code (Wen et al., 2023). However, recent works have demonstrated that directly interacting with the visual form of GUIs leads to better performance (Hu et al., 2024a). Consequently, MLLM-based GUI agents have been proposed, leveraging visual perception alongside language understanding.

Several representative systems have pioneered this area. ILuvUI (Jiang et al., 2023) fine-tuned LLaVA to enhance general GUI comprehension, while AppAgent (Zhang et al., 2023) explored mobile app usage through autonomous interactions. CogAgent (Hong et al., 2024) introduced high-resolution encoders to better capture UI detail, and Ferret-UI-anyres (You et al., 2025) supported flexible screen resolutions to handle diverse device settings.

More recent works have introduced modular and lightweight architectures aimed at improving generalization and deployment efficiency. InfiGUIAgent (Liu et al., 2025) proposed a two-stage approach, combining general pretraining on grounding and QA tasks with synthetic fine-tuning for hierarchical planning and reasoning. UI-TARS (Qin et al., 2025b) extended this by using a unified vision-language interface across mobile, web, and desktop environments, incorporating reflection and milestone tracking mechanisms to boost task success rates. In parallel, AgentS2 (Agashe et al., 2025) adopted a generalist-specialist framework, decoupling high-level reasoning from domain-specific grounding modules and enabling long-horizon planning with Mixture of Grounding mechanisms.

In terms of input, recent agents prioritize screenshot-level visual understanding, optionally enhanced with layout or OCR-based textual cues. Techniques such as set-of-mark prompting (Yang et al., 2023) and chain-of-action reasoning (Pan et al., 2024) have been employed to improve grounding accuracy and task planning. To further improve interaction efficiency, agents such as UI-R1 (Lu et al., 2025), GUI-R1 (Xia & Luo, 2025) replace large-scale supervision with rule-based reinforcement learning, achieving competitive performance with minimal expert data.

Moreover, to support real-world usability, newer agents are tested on increasingly complex environments. UI-TARS and AgentS2 report strong performance on OSWorld and AndroidWorld benchmarks, showing robust cross-platform generalization. GUI-Xplore (Sun et al., 2025) further introduces a one-shot adaptation setting, encouraging agents to build structural UI maps via autonomous exploration before task execution.

## 3 THE ACTOR2REASONER PARADIGM

Our work is built upon **Actor2Reasoner**, a two-stage hierarchical training paradigm designed to resolve the structural mismatch previously identified. Grounded in the principle of *Endow First, Internalize Later*, our paradigm decomposes the complex challenge of training a GUI agent into two orthogonal yet complementary stages: Cognitive Endowment and Policy Internalization. The process begins with an MLLM already pre-trained on fundamental GUI capabilities, such as element grounding and action execution. Our paradigm then orchestrates its evolution, first transforming it into a *Capable Actor* with robust GUI-centric reasoning patterns, and subsequently refining it into a *Deliberative Reasoner* that can internalize these patterns into an optimal decision-making policy.

### 3.1 STAGE 1: COGNITIVE ENDOWMENT

To instill a robust behavioral prior, which is a prerequisite for efficient policy learning, the first stage of our paradigm focuses on cognitive endowment. By decoupling the acquisition of foundational reasoning skills from goal-oriented policy optimization, we formulate a more tractable learning problem for the subsequent RL stage. This endowment is achieved through a specialized form of knowledge distillation. Specifically, we leverage a powerful teacher model to retroactively synthesize targeted Chain-of-Thought (CoT) rationales that serve as explicit supervision. Given a state (screenshot and instruction) and a ground-truth action, the teacher model is prompted to reverse-engineer the ideal thought process that leads to that action. This synthesis process can be formulated as:

$$z_i = \mathcal{M}_{\text{teacher}}(x_i, a_i) \tag{1}$$

where for a given sample $i$, $x_i$ is the input state, $a_i$ is the ground-truth action, and $\mathcal{M}_{\text{teacher}}$ is the teacher model. The output $z_i$ is the synthesized CoT, representing the structured thought.

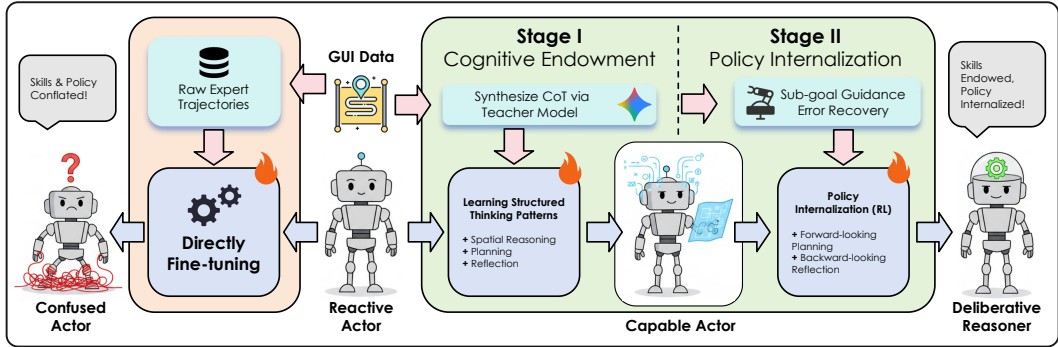

Figure 1: An illustration of our proposed **Actor2Reasoner** paradigm in contrast to the conventional monolithic paradigm. **Left:** The monolithic paradigm fine-tunes a base agent on raw expert trajectories, conflating the learning of foundational skills with policy optimization. This leads to a *Confused Actor* trapped by a structural mismatch, unable to learn effectively. **Right:** Our **Actor2Reasoner** paradigm decouples these two objectives. In **Stage I (Cognitive Endowment)**, we process raw data via a teacher model to synthesize structured thinking patterns, which are then used to endow a *Reactive Actor* with key skills (e.g., spatial reasoning, planning), transforming it into a *Capable Actor*. In **Stage II (Policy Internalization)**, the *Capable Actor* is trained with RL in scenarios featuring sub-goal guidance and error recovery. This internalizes the endowed skills into a robust policy, culminating in a *Deliberative Reasoner* that succeeds because its foundational skills are endowed before its policy is internalized.

The synthesis process is meticulously designed to be task-aware. For different categories of GUI tasks, we prompt the teacher model to generate CoT with specific, structured reasoning patterns. To enhance the quality and fidelity of the synthesized thoughts, particularly for spatial reasoning which demands high precision, we supply the teacher model with screenshots where key interactive elements are programmatically marked with their coordinates. This significantly reduces the teacher model's visual perception burden and allows it to focus on generating high-quality reasoning steps. The synthesis is designed to instill three core reasoning patterns critical for GUI interaction:

- **Spatial Reasoning:** For localization-centric tasks (e.g., grounding), the synthesized CoT is structured to follow a "global-to-local" analytical flow. This forces the agent to learn to reason about element positions in a structured manner, rather than relying on simple pattern matching. For instance: *"The 'Settings' icon is located in the top-right quadrant of the screen, near the battery indicator."*
- **Goal Decomposition:** For trajectory-based tasks, the CoT explicitly breaks down a high-level user instruction into an immediate, actionable sub-goal. This teaches the agent to formulate a concrete, executable intention for the current step, which is a foundational skill for planning. For example: *"The user instruction is 'Book a ride to the airport'. The immediate sub-goal is therefore to input 'airport' into the destination text field."*
- **Reflection:** For trajectory-based tasks, the CoT also includes a reflective step to verify the outcome of the previous action. This lays the cognitive groundwork for the error recovery capabilities that will be honed in the second stage. For instance: *"The last action was to click the login button at top-right. The screen has now transitioned to the login page as expected. The next step is to..."*

To further ensure the correctness and utility of the synthesized CoT, we introduce a verification-filtering step. For each generated CoT $z_i$, we concatenate it with the original instruction and feed it to our agent model to generate a corresponding action $a_i'$. We retain the data sample $(x_i, z_i, a_i)$ for training only if the predicted action $a_i'$ matches the ground-truth action $a_i$. This filtering process guarantees that the training data consists of high-fidelity reasoning chains that are demonstrably effective in guiding the agent to produce the correct action, thus mitigating potential noise from the teacher model.

The agent is then trained via SFT on this curated and verified dataset of state-thought-action triplets. The model is thus trained not merely to replicate actions, but to first generate a structured thought

process. This explicitly teaches the model to *think before it acts* by internalizing the specific *thinking patterns* detailed above. The objective is to predict the composite sequence $y_i$, which is the concatenation of the thought $z_i$ and the action $a_i$, optimizing a standard cross-entropy loss. This process effectively distills the structured reasoning from the teacher model into our agent, creating the *Capable Actor*.

### 3.2 STAGE 2: POLICY INTERNALIZATION

With the *Capable Actor* forged in Stage 1, the agent now possesses a robust behavioral prior grounded in key GUI-centric thinking patterns. The objective of Stage 2, Policy Internalization, is to evolve this actor into a *Deliberative Reasoner*. We employ Reinforcement Learning (RL) as the mechanism to refine and apply these endowed capabilities in situated contexts. This is achieved by immersing the agent in a curriculum of tasks specifically designed to internalize sophisticated thinking patterns, such as forward-looking planning and backward-looking reflection, into a robust decision-making policy.

#### 3.2.1 STRENGTHENING SPATIAL REASONING VIA GROUNDING TASKS

Precise visual grounding is a cornerstone of competent GUI interaction. Unlike agents with access to structured view hierarchies, our MLLM-based agent operates on raw pixels, making the ability to accurately locate and interact with specified elements a critical prerequisite for any downstream task. To hone this skill, we formulate a dedicated grounding task and design a composite reward signal that provides both sparse and dense feedback.

- **Point-based Reward** ($\mathcal{R}_{\textbf{point}}$): This provides an intuitive, direct signal for interaction success. The agent, tasked with predicting an interaction coordinate $(x, y)$, receives a positive reward if its prediction falls within the ground-truth bounding box of the target element. While effective, this reward is sparse and provides no gradient for improving precision within the target area.
- **Bounding Box Reward** ($\mathcal{R}_{\textbf{bbox}}$): To provide a denser learning signal, we also train the agent to directly predict the target element's bounding box, $B_{\text{pred}}$. This encourages the agent to better comprehend the element's spatial extent and boundaries, which in turn facilitates more precise interaction. The reward is formulated as the Intersection over Union (IoU):

$$\mathcal{R}_{\text{bbox}} = \text{IoU}(B_{\text{pred}}, B_{\text{gt}}) \tag{2}$$

where $B_{\text{gt}}$ is the ground-truth bounding box. This combination of rewards ensures the agent learns not only to hit the target but to do so with a comprehensive understanding of its location and shape.

#### 3.2.2 FORWARD-LOOKING GUIDANCE IN TRAJECTORY TASKS

For multi-step trajectory tasks, a core challenge is the temporal credit assignment problem, where a final outcome is the result of a long sequence of actions. To provide more immediate and meaningful supervision, we introduce a forward-looking guidance mechanism that directly rewards the agent's internal thought process. This mechanism leverages the goal decomposition capability endowed in Stage 1. Specifically, we reward the agent for correctly predicting the immediate sub-goal for the current step, thus aligning its internal planning with the task's ground truth. The sub-goal reward is defined as:

$$\mathcal{R}_{\text{sub}} = \text{sim}(g_{\text{pred}}, g_{\text{gt}}) \tag{3}$$

where $g_{\text{pred}}$ is the agent's predicted sub-goal and $g_{\text{gt}}$ is the ground-truth sub-goal from our curated dataset. The function $\text{sim}(\cdot)$ computes the semantic similarity between the two, providing a dense signal that guides the agent's planning process at each step.

#### 3.2.3 REFLECTIVE CORRECTION VIA ERROR RECOVERY SCENARIOS

Complementing the forward-looking guidance is our backward-looking correction mechanism, designed to address a critical flaw in offline datasets: the absence of negative examples. Agents trained solely on expert trajectories are brittle and often fail to recover from their own mistakes. To build resilience, we explicitly teach a recovery policy by constructing targeted error recovery scenarios.

Let a correct trajectory be denoted as $\tau = (s_0, a_0, s_1, \ldots, s_T)$. At a given state $s_t$, if the agent predicts an erroneous action $a'_t \neq a_t$, it transitions to an unexpected error state $s'_{t+1}$. Our goal is

to teach the agent an effective three-step recovery process: identify the error, escape the error state, and recover to the correct trajectory. This is encouraged through two dedicated reward signals:

- **Escape Reward** ($\mathcal{R}_{\text{esc}}$): In the error state $s'_{t+1}$, the agent is rewarded for executing a designated "escape" action (e.g., a 'back' operation) that successfully returns it to the previous known-good state $s_t$.
- **Back-on-Track Reward** ($\mathcal{R}_{\text{rec}}$): Upon returning to state $s_t$, the agent receives a further reward for then executing the correct action $a_t$, successfully getting back on the original trajectory $\tau$.

The synergy between our forward-looking sub-goal guidance and backward-looking error correction mechanisms fosters a robust agent that can not only proactively plan its next move but also reactively recover from unforeseen errors.

### 3.3 TRAINING FORMULATION

#### 3.3.1 REWARD CALCULATION

To unify the diverse learning signals from grounding, planning, and error recovery, we formulate a comprehensive reward function. The total reward $\mathcal{R}_{\text{total}}$ for any given action is composed of two independent components: a format reward and an accuracy reward.

$$\mathcal{R}_{\text{total}} = \mathcal{R}_{\text{format}} + \mathcal{R}_{\text{accuracy}} \tag{4}$$

The **Format Reward** ($\mathcal{R}_{\text{format}}$) is a binary signal that returns $+1$ if the agent's predicted action conforms to the predefined action format, and $0$ otherwise. This encourages the model to generate syntactically valid outputs.

The **Accuracy Reward** ($\mathcal{R}_{\text{accuracy}}$), normalized to the range $[-1, 1]$, measures the semantic correctness and quality of the action. Its composition is task-dependent:

- For **Grounding Tasks**, the accuracy reward is directly assigned from one of the previously defined spatial reasoning rewards. Depending on the specific task objective, $\mathcal{R}_{\text{accuracy}}$ is set to either the point-based reward $\mathcal{R}_{\text{point}}$ or the bounding box reward $\mathcal{R}_{\text{bbox}}$ (§3.2).
- For **Trajectory Tasks**, to provide a denser signal, the accuracy reward is calculated as the sum of three normalized sub-rewards, each ranging from $[0, 1]$:

$$\mathcal{R}_{\text{accuracy}} = \mathcal{R}_{\text{type}} + \mathcal{R}_{\text{param}} + \mathcal{R}_{\text{sub}} \tag{5}$$

Here, $\mathcal{R}_{\text{type}}$ rewards the prediction of the correct action type (e.g., 'click', 'type'), $\mathcal{R}_{\text{param}}$ rewards the prediction of the correct action parameters (e.g., coordinates, input text), and $\mathcal{R}_{\text{sub}}$ is the sub-goal similarity reward defined in Eq. 3. This multi-component reward provides fine-grained feedback on different facets of the agent's action generation process.

#### 3.3.2 DATA FILTERING FOR EFFICIENT TRAINING

To maximize the efficiency of the RL phase, particularly for learning from rare but critical events like error recovery, we employ a data filtering strategy to focus on the most informative samples. We first use the *Capable Actor* from Stage 1 to perform $K$ rollouts for each sample in our dataset. We calculate the success rate $p$ for each trajectory. For the RL training, we exclusively use the subset of data $\mathcal{D}_{\text{high-potential}}$ where the success rate is neither always perfect nor always failing. These samples reside at the agent's current "learning frontier" and thus provide the richest signal for policy improvement.

#### 3.3.3 OPTIMIZATION ALGORITHM

We utilize Reinforcement Learning with Verifiable Rewards (RLVR) as our optimization algorithm. The policy $\pi_\theta$ is trained to maximize the expected reward, with the objective function given by:

$$\theta^* = \arg\max_\theta \mathbb{E}_{c \sim \mathcal{D}, a \sim \pi_\theta(\cdot|c)}[R(a, B)] \tag{6}$$

where $c$ is the context, $a$ is the action sampled from the policy, and $R(a, B)$ is our verifiable reward function.

# 4 EXPERIMENTS

## 4.1 EXPERIMENT SETUP

### 4.1.1 BENCHMARKS AND METRICS

Our empirical evaluation is structured around two fundamental pillars of GUI agent capabilities: visual grounding and agentic task execution.

**Benchmarks.** For assessing **visual grounding**, we utilize **ScreenSpot-V2** for its broad coverage of common interactive elements, and its more challenging successor, **ScreenSpot-Pro**, to evaluate performance on high-resolution screens from professional software. For evaluating **agentic task execution**, we employ the **AndroidControl** benchmark to assess performance in multi-step tasks within real-world mobile applications. We also use **AndroidWorld** to further evaluate the agent's generalization capabilities in executing complex, long-horizon tasks based on high-level instructions.

**Metrics.** For the grounding benchmarks, our primary metric is **Accuracy**, where a prediction is deemed correct if the predicted coordinate falls within the ground-truth bounding box of the target element. For the **AndroidControl** benchmark, we adopt a suite of three fine-grained Success Rate (SR) metrics to provide a comprehensive performance profile: **Type SR** (whether the action type is correct), **Grounding SR** (the click precision for interaction-based actions), and **Overall Step SR** (whether both type and parameters are completely correct).

### 4.1.2 BASELINES

To comprehensively evaluate the effectiveness of our proposed paradigm, we compare InfiGUI-R1 against a diverse set of recent, state-of-the-art GUI agents. These baseline methods cover a range of model architectures, sizes, and training paradigms, representing the latest advancements in the field.

### 4.1.3 IMPLEMENTATION DETAILS

Our InfiGUI-R1 is built upon the open-source Qwen2.5-VL-3B-Instruct model. For the Chain-of-Thought (CoT) synthesis in Stage 1, we utilize Gemini-2.5-Flash as the teacher model. In the policy internalization stage, we employ the **RLOO (Reinforcement Learning with Leave-One-Out)** algorithm (Ahmadian et al., 2024). RLOO effectively reduces the variance of policy gradient estimates by using the average reward of other samples within the same batch as a baseline, thereby obviating the need for a separate critic network.

### 4.1.4 TRAINING DETAILS

**Data.** The training process for both stages utilizes a consistently structured dataset. The SFT in Stage 1 is conducted on a curated set of approximately 3k samples. The RL in Stage 2 uses a larger dataset of approximately 44k samples. For both stages, the dataset is composed of a balanced 50/50 split between grounding tasks (collated from various public GUI datasets, including Widget Caption, OmniAct, and GUICourse) and agent tasks (from the AndroidControl training set) to ensure balanced development of both core capabilities.

**Hyperparameters.** For the SFT stage, we set the batch size to 32 and the learning rate to 4e-6 with a cosine schedule, training for 2 epochs. During this stage, both the vision encoder and the vision projector of the backbone model were frozen. For the RL stage, the full model parameters are trainable, and we use a learning rate of 1e-6, a batch size of 128, and train for 2 epochs. For the RLOO algorithm, the number of rollouts was set to $n = 8$. All experiments were conducted on 8 NVIDIA H800 GPUs.

## 4.2 MAIN RESULTS

Our agent, **InfiGUI-R1**, substantiates the effectiveness of the Actor2Reasoner paradigm by achieving state-of-the-art or highly competitive performance across a suite of challenging benchmarks.

Table 1: Performance comparison on the **ScreenSpot-V2** benchmark. Best and second-best scores are shown in **bold** and underlined, respectively.

| Model | Mobile | | Desktop | | Web | | Avg. |
|---|---|---|---|---|---|---|---|
| | Text | Icon | Text | Icon | Text | Icon | |
| SeeClick (Cheng et al., 2024) | 78.4 | 50.7 | 70.1 | 29.3 | 55.2 | 32.5 | 55.1 |
| OS-Atlas-Base-7B (Wu et al., 2024) | 95.2 | 75.8 | 90.7 | 63.6 | 90.6 | 77.3 | 85.1 |
| UI-TARS-7B (Qin et al., 2025a) | 96.9 | **89.1** | 95.4 | 85.0 | 93.6 | 85.2 | 91.6 |
| UI-TARS-72B (Qin et al., 2025a) | 94.8 | 86.3 | 91.2 | **87.9** | 91.5 | 87.7 | 90.3 |
| Qwen2.5-VL-3B (Bai et al., 2025) | 93.4 | 73.5 | 88.1 | 58.6 | 88.0 | 71.4 | 80.9 |
| Qwen2.5-VL-7B (Bai et al., 2025) | 97.6 | 87.2 | 90.2 | 74.2 | 93.2 | 81.3 | 88.8 |
| **InfiGUI-R1-3B (Ours)** | **99.0** | 85.8 | 95.9 | 77.9 | 92.7 | 75.9 | 89.5 |
| *w/o Cognitive Endowment* | **99.0** | 84.4 | 94.3 | 74.3 | 93.2 | 78.8 | 88.8 |
| *w/o Policy Internalization* | 96.6 | 82.5 | 87.6 | 70.7 | 83.8 | 70.9 | 82.0 |

Table 2: Performance comparison on the **ScreenSpot-Pro** benchmark. Best and second-best scores are shown in **bold** and underlined, respectively.

| Model | CAD | | Dev. | | Creative | | Scientific | | Office | | OS | | Avg. |
|---|---|---|---|---|---|---|---|---|---|---|---|---|---|
| | Text | Icon | Text | Icon | Text | Icon | Text | Icon | Text | Icon | Text | Icon | |
| GPT-4o (Hurst et al., 2024) | 2.0 | 0.0 | 1.3 | 0.0 | 1.0 | 0.0 | 2.1 | 0.0 | 1.1 | 0.0 | 0.0 | 0.0 | 0.8 |
| Claude Comp. Use (Anthropic, 2024) | 14.5 | 3.7 | 22.0 | 3.9 | 25.9 | 3.4 | 33.9 | 15.8 | 30.1 | 16.3 | 11.0 | 4.5 | 17.1 |
| SeeClick (Cheng et al., 2024) | 2.5 | 0.0 | 0.6 | 0.0 | 1.0 | 0.0 | 3.5 | 0.0 | 1.1 | 0.0 | 2.8 | 0.0 | 1.1 |
| Qwen2-VL-7B (Wang et al., 2024) | 0.5 | 0.0 | 2.6 | 0.0 | 1.5 | 0.0 | 6.3 | 0.0 | 3.4 | 1.9 | 0.9 | 0.0 | 1.6 |
| CogAgent-18B (Hong et al., 2024) | 7.1 | 3.1 | 14.9 | 0.7 | 9.6 | 0.0 | 22.2 | 1.8 | 13.0 | 0.0 | 5.6 | 0.0 | 7.7 |
| UI-R1-3B (Lu et al., 2025) | 11.2 | 6.3 | 22.7 | 4.1 | 27.3 | 3.5 | 42.4 | 11.8 | 32.2 | 11.3 | 13.1 | 4.5 | 17.8 |
| ZonUI-3B (Hsieh et al., 2025) | 31.9 | 15.6 | 24.6 | 6.2 | 40.9 | 7.6 | 54.8 | 18.1 | 57.0 | 26.4 | 19.6 | 7.8 | 28.7 |
| GUI-R1-7B (Xia & Luo, 2025) | 23.9 | 6.3 | 49.4 | 4.8 | 38.9 | 8.4 | 55.6 | 11.8 | 58.7 | 26.4 | 42.1 | 16.9 | 31.0 |
| UI-TARS-7B (Qin et al., 2025b) | 20.8 | 9.4 | 58.4 | 12.4 | 50.0 | 9.1 | 63.9 | **31.8** | 63.3 | 20.8 | 30.8 | 16.9 | 35.7 |
| **InfiGUI-R1-3B (Ours)** | 45.2 | **20.3** | **67.5** | **17.2** | **54.0** | 14.7 | **65.3** | **31.8** | **69.5** | 24.5 | 42.1 | **18.0** | **43.3** |
| *w/o Cognitive Endowment* | **49.7** | 12.5 | 58.4 | 15.2 | 50.0 | **15.4** | 63.2 | 27.3 | 66.1 | **32.1** | **46.7** | 15.7 | 41.6 |
| *w/o Policy Internalization* | 30.5 | 10.9 | 50.0 | 9.7 | 39.9 | 7.0 | 56.9 | 18.2 | 59.3 | 24.5 | 40.2 | 9.0 | 32.8 |

**Visual Grounding Performance.** InfiGUI-R1 demonstrates robust grounding capabilities, particularly on the complex **ScreenSpot-Pro** benchmark (Table 2), where our 3B model establishes a new state-of-the-art with a 43.3% average accuracy. This result surpasses all baselines, including significantly larger models, and highlights the efficacy of our Stage 1 Cognitive Endowment in instilling superior spatial reasoning abilities, especially in icon-dense professional software environments.

**Agentic Task Performance.** In the realm of complex, multi-step tasks, InfiGUI-R1 showcases remarkable policy execution. On the **AndroidControl** benchmark (Table 3), our agent achieves a state-of-the-art 71.1% success rate on the High-level setting, which requires autonomous planning without explicit step-by-step guidance. This strong performance, further corroborated by results on **AndroidWorld** (Table 4), provides strong evidence that our Stage 2 Policy Internalization effectively translates endowed cognitive skills into a robust, goal-oriented policy.

## 4.3 ABLATION STUDIES

We conduct a comprehensive ablation study to validate our architectural claims and dissect the contributions of each component. The results consistently confirm the necessity of our two-stage design and the synergistic effect between its parts.

**The Necessity of the Two-Stage Paradigm.** Our core hypothesis is that decoupling skill endowment and policy internalization is critical. As shown across our agentic benchmarks (Table 3 and 4), removing the **Policy Internalization** stage (*w/o Policy Internalization*) causes the most significant performance drop (e.g., a 50% relative drop on AndroidWorld). This highlights the necessity of RL for absorbing the distilled skills into an executable policy. Conversely, removing the **Cognitive Endowment** stage (*w/o Cognitive Endowment*) also consistently degrades performance, confirming our premise that RL struggles to learn efficiently without a strong behavioral prior of key reasoning

Table 3: Performance comparison of different agent models on AndroidControl benchmarks. SR stands for Success Rate. Results marked in **bold** represent the best performance, and those underlined indicate the second-best performance.

| Model | AndroidControl-Low | | | AndroidControl-High | | |
|---|---|---|---|---|---|---|
| | Type | Grounding | SR | Type | Grounding | SR |
| Claude* | 74.3 | 0.0 | 19.4 | 63.7 | 0.0 | 12.5 |
| GPT-4o | 74.3 | 0.0 | 19.4 | 66.3 | 0.0 | 20.8 |
| Aria-UI (Yang et al., 2024) | – | 87.7 | 67.3 | – | 43.2 | 10.2 |
| OS-Atlas-4B (Wu et al., 2024) | 91.9 | 83.8 | 80.6 | **84.7** | 73.8 | 67.5 |
| Aguvis-7B (Xu et al., 2025) | – | – | 80.5 | – | – | 61.5 |
| Aguvis-72B (Xu et al., 2025) | – | – | 84.4 | – | – | 66.4 |
| UI-R1 (Lu et al., 2025) | 94.3 | 82.6 | - | - | - | - |
| GUI-R1-3B (Xia & Luo, 2025) | - | - | - | 58.0 | 56.2 | 46.6 |
| GUI-R1-7B (Xia & Luo, 2025) | - | - | - | 71.6 | 65.6 | 51.7 |
| UI-TARS-2B (Qin et al., 2025a) | **98.1** | 87.3 | 89.3 | 81.2 | **78.4** | 68.9 |
| **InfiGUI-R1-3B (Ours)** | 95.8 | 92.6 | 91.8 | 82.1 | 73.5 | **71.1** |
| *w/o Cognitive Endowment* | 95.8 | 92.6 | 91.2 | 80.9 | 72.7 | 69.5 |
| *w/o Policy Internalization* | 96.0 | 91.6 | 90.6 | 79.4 | 66.0 | 63.8 |
| *w/o Sub-goal Guidance* | 95.5 | 93.4 | 91.0 | 80.4 | 71.5 | 68.0 |
| *w/o Error Recovery* | 96.0 | **93.5** | **92.0** | 81.2 | 73.6 | 70.0 |

Table 4: Ablation on **AndroidWorld**. We report Task Success (↑) over 116 tasks. Cognitive Endowment: Stage 1 spatial reasoning / CoT distillation. Policy Internalization: Stage 2 RLOO-based RL. Sub-goal Guidance and Error Recovery are Stage 2 objectives that facilitate planning and self-correction.

| Method | Success # / 116 (↑) | Success % (↑) |
|---|---|---|
| InfiGUI-R1-3B (full) | **20** | **17.2** |
| w/o Cognitive Endowment | 17 | 14.7 |
| w/o Policy Internalization | 10 | 8.6 |
| w/o Sub-goal Guidance | 14 | 12.1 |
| w/o Error Recovery | 15 | 12.9 |

skills. The full model consistently outperforms both ablations, validating the powerful synergistic effect of our two-stage approach.

**The Impact of Stage 2 Mechanisms.** We further ablated the two core mechanisms within our Policy Internalization stage. Removing the forward-looking **Sub-goal Guidance** or the backward-looking **Error Recovery** each leads to a significant performance drop across agentic tasks. These results empirically validate their indispensable roles in fostering robust planning and correction capabilities. In summary, our ablations confirm that Stage 1 grants the essential priors, and Stage 2, with all its components, effectively internalizes them into a robust, self-correcting policy.

## 5 CONCLUSION

In this work, we identified a structural mismatch in monolithic training paradigms for GUI agents, which conflate the learning of foundational reasoning skills with policy optimization. To resolve this, we proposed **Actor2Reasoner**, a two-stage hierarchical paradigm that first endows an agent with key cognitive abilities before internalizing them into a robust policy. Our instantiation, **InfiGUI-R1**, achieves state-of-the-art performance on challenging benchmarks, empirically validating our architectural principles. Our findings suggest that decoupling skill endowment from policy internalization offers a more principled and effective path toward developing the next generation of sophisticated and resilient GUI agents.

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

# A  APPENDIX

## THE USE OF LARGE LANGUAGE MODELS (LLMS)

LLMs were used exclusively as writing assistance tools in preparing this manuscript. Specifically, we employed LLMs for grammar checking. All research ideation, experimental design, analysis, and scientific conclusions are entirely the work of the authors. The LLMs played no role in the conception of research questions, methodology development, or interpretation of results. Authors take full responsibility for all content in this paper, including any text refined with LLM assistance.

