# OpenReview forum: "InfiGUI-R1: Advancing Multimodal GUI Agents from Reactive Actors to Deliberative Reasoners"
_ICLR.cc/2026/Conference — Submitted to ICLR 2026_

### Official Review · Reviewer_BcjR · 2025-10-17

**Soundness:** 3
**Presentation:** 3
**Contribution:** 2
**Rating:** 6
**Confidence:** 3

**Summary:**

The authors introduce InfiGUI-R1, a vision-language-action model to interact with user interfaces and accomplish tasks. The model is trained in a 2-stage training pipeline, first with supervised fine tuning over a clean action dataset, and second with reinforcement learning over trajectories that are “just hard enough” so as to remain on the learning frontier of the model. The model is trained with several different objectives depending on the task at hand, varying from bounding-box IoU to sub-goal matching to action prediction. As part of the learning process, the authors also propose a sub-goal prediction/matching task, and an error correction task. These are both claimed to help improve final performance.

InfiGUI-R1 is compared to much larger models on ScreenSpot benchmarks for grounding performance, and on AndroidControl and AndroidWorld benchmarks for interaction performance. The model achieves comparable or superior performance when compared to baselines, despite being smaller. The authors also conduct an ablation study in which they remove either their SFT phase or their RL phase, and they show that both are necessary to the success of the final model. Ablations on the subgoal and error recover tasks seem to indicate that they are less important to the overall success of the pipeline, with the “no error recovery” model even outperforming the full model along some metrics.

**Strengths:**

+ The final performance of the InfiGUI-R1 model performs very well, achieving superior performance than larger models
+ The proposed pipeline is intuitive and reflects many sensible design decisions, and would be easy to replicate by other researchers in their own workflows.
+ The experiments evaluate InfiGUI-R1 along several important axes, showing results for different GUI instantiations (mobile/web/desktop), grounding performance on different tasks/applications, and task success on different benchmarks. These greatly help to illustrate the strengths of the model, and to contextualize it in the scope of related work.

**Weaknesses:**

- **Novelty**: SFT into RLVR is not a particularly new training paradigm. Every major LLM since ChatGPT has featured some combination of these two training paradigms in this order, and so the insight to apply SFT and then RLVR for GUI navigation is not particularly novel or insightful. In fact, one of the baselines in the paper, UI-TARS, employs an SFT-> DPO training pipeline, which is of course very similar. Similarly, many vision-language-action models in the robotics community have already demonstrated the success of SFT -> RLHF for task learning with VLMs.
- Clarity: Elements of the paper are left unclear. For example, the authors introduce either a point-based reward or a bounding-box reward for certain UI tasks, but then there is no comparison between these two, or discussion on when to use each. This does not help future work to build on anything that the authors may have learned from experimenting with these rewards for UI navigation or grounding. Similarly, elements of the paper are left slightly unclear, such as how the Accuracy reward is normalized to [-1, 1], but sometimes it is set to the sum of 3 rewards ranging from [0, 1]… So it should range from [0, 3]. Is this reward then re-normalized to [-1, 1]? If so, is it a valid learning signal?
- (Minor) Formatting: nearly all of the opening quotes are backwards, and some of the closing quotes are backwards. The paper could use a formatting check.

While does not seem to be anything technically wrong with the paper and the results are fairly strong, the contributions are minimal. The techniques being used, like data filtering, SFT -> RL, and RLOO, are all well established in the literature at this point, and so there is not much new being added. Nonetheless, the results are strong, and so I lean towards weak accept.

**Questions:**

The error recovery component seems to be both very trivial to solve (hit a back button) and also somewhat damaging to performance (as evidenced by the occasionally higher performance of the InfiGUI method that doesn’t use error recovery in training). Does this suggest that that task is bad for helping to learn general GUI navigation? Or is there something else that explains why the method does better without error recovery?

---

> ### Author Response · Authors · 2025-12-03
> **[Part 1/2] Response to Reviewer BcjR**
>
> We sincerely thank the reviewer for the constructive and detailed feedback. We have carefully examined each concern and address every point in detail below.
>
> ## Weakness 1
> > **Novelty: SFT into RLVR is not a particularly new training paradigm. Every major LLM since ChatGPT has featured some combination of these two training paradigms in this order, and so the insight to apply SFT and then RLVR for GUI navigation is not particularly novel or insightful. In fact, one of the baselines in the paper, UI-TARS, employs an SFT-> DPO training pipeline, which is of course very similar. Similarly, many vision-language-action models in the robotics community have already demonstrated the success of SFT -> RLHF for task learning with VLMs.**
>
> Our contribution is **NOT** the mere existence of a SFL-RL pipeline, but focused on answering two key questions specific to the GUI domain: **How should SFT provide the most effective prior for RL?** and **How should RL's reward structure be designed to build upon the SFT prior?**  For instance, part of our novel designs is:
>
> 1.  **Enriched SFT Prior (SFT Stage, Section 3.1):** We specifically engineered the SFT phase to teach **three fundamental GUI-centric reasoning patterns**—Spatial Reasoning, Goal Decomposition, and Reflection—by curating and filtering the dataset based on these concepts. This deliberate design ensures the SFT model acquires the essential **atomic GUI skills** necessary for the RL agent to effectively explore and succeed in complex, multi-step environments.
>
> 2.  **Dense, Forward-Looking Guidance (RL Stage, Section 3.2.2):** We introduce **Forward-Looking Guidance** as a core RL mechanism to directly address the severe **temporal credit assignment problem** inherent in long-horizon GUI tasks. We utilize the **Goal Decomposition capability** learned in Stage 1 to provide a **dense reward signal** based on the agent's *internal planning* process:
> > The Sub-goal Reward ($R_{\text{sub}}$) is defined by the semantic similarity between the agent's predicted immediate sub-goal ($g_{\text{pred}}$) and the ground-truth sub-goal ($g_{\text{gt}}$).     This $R_{\text{sub}}$ provides **immediate, dense supervision** at every step, guiding the agent's planning process and ensuring better alignment between its internal reasoning and the task objective, which is far more nuanced than a typical sparse final success reward.
>
> Our superior empirical results, even against larger models and similar baselines like UI-TARS (which uses SFT $\rightarrow$ DPO), validate the necessity and effectiveness of these specific, tailored innovations in data and reward design within the established SFT $\rightarrow$ RL framework.
>
> ## Weakness 2
> > **Clarity: Elements of the paper are left unclear. For example, the authors introduce either a point-based reward or a bounding-box reward for certain UI tasks, but then there is no comparison between these two, or discussion on when to use each. This does not help future work to build on anything that the authors may have learned from experimenting with these rewards for UI navigation or grounding. Similarly, elements of the paper are left slightly unclear, such as how the Accuracy reward is normalized to [-1, 1], but sometimes it is set to the sum of 3 rewards ranging from [0, 1]… So it should range from [0, 3]. Is this reward then re-normalized to [-1, 1]? If so, is it a valid learning signal?**
>
> ### 1. Choice Between $R_{point}$ and $R_{bbox}$
>
> The choice between the point-based reward ($R_{point}$) and the bounding box reward ($R_{bbox}$) is **task-dependent** and determined by the required action output format, **as already stated in Section 3.3.1**.
>
> * **$R_{point}$** is used for **Interactive Actions** (e.g., `click`, `long press`) that fundamentally require the prediction of a single interaction coordinate $(x, y)$. The reward signals whether the predicted point falls within the ground-truth target area.
>
> * **$R_{bbox}$** is used for **Grounding/Information Extraction Tasks** (e.g., in the ScreenSpot benchmark) that require the model to **output the full target element's bounding box** ($B_{pred}$). Here, we directly optimize IoU to hone spatial understanding precision.
>
>
> ### 2. Accuracy Reward Normalization
>
> Regarding the normalization of the accuracy reward ($R_{accuracy}$) for trajectory tasks, the reviewer is correct that the raw sum of the three sub-components ($S = R_{type} + R_{param} + R_{sub}$) ranges from $[0, 3]$.
> We map this sum to the $[-1, 1]$ interval using a standard **linear scaling function**:
> $$ R_{accuracy} = \frac{2}{3} \cdot S - 1 $$
> *   **Minimum:** If the model gets everything wrong ($S=0$), the reward is $-1$.
> *   **Maximum:** If the model is perfect ($S=3$), the reward is $+1$.
>
> Generally, we will refine our scripts to ensure greater clarity and transparency in the final version.

---

> ### Author Response · Authors · 2025-12-03
> **[Part 2/2] Response to Reviewer BcjR**
>
> ## Weakness 3
> > (Minor) Formatting: nearly all of the opening quotes are backwards, and some of the closing quotes are backwards. The paper could use a formatting check.
>
> We thank the reviewer for pointing out the typographical issue with the quotation marks (opening and closing quotes). We acknowledge this minor formatting error and assure the reviewer that we will conduct a thorough formatting check on the final manuscript to correct this and any other minor issues.
>
> ## Question 1
> > The error recovery component seems to be both very trivial to solve (hit a back button) and also somewhat damaging to performance (as evidenced by the occasionally higher performance of the InfiGUI method that doesn’t use error recovery in training). Does this suggest that that task is bad for helping to learn general GUI navigation? Or is there something else that explains why the method does better without error recovery?
>
> 1. The experimental evidence clearly shows that the absence of Error Recovery training **severely impairs performance** on the complex, long-horizon tasks found in the **AndroidWorld** benchmark (Table 4):
>
> * **AndroidWorld: $17.2 \rightarrow 12.9$** (When Removing No Error Recovery)
>
> This dramatic drop of **4.3 points** demonstrates that explicit error recovery training is crucial for building robust decision-making and resilience, especially when tasks are complex and mistakes are common.
>
> Although we observed a marginal, minor increase in performance on the simpler **AndroidControl-Low** dataset ($91.8 \rightarrow 92.0$) when removing Error Recovery, the performance on the more challenging **AndroidControl-High** subset also saw a decline ($71.1 \rightarrow 70.0$).
>
> Therefore, the **gain provided by Error Recovery (as seen in the critical AndroidWorld benchmark) far outweighs the minimal or negligible negative effects** observed elsewhere.
>
> 2. Regarding the perceived simplicity of the "back" action, our case analysis reveals that for smaller agents operating solely on raw pixels, even this simple recovery step—which requires identifying the error state and choosing a corrective *escape* action—is **non-trivial to learn without explicit supervision**. Without this training, the small model often gets perpetually trapped in unexpected error states, which is exactly what the Error Recovery task is designed to prevent.

---

### Official Review · Reviewer_sQ4b · 2025-10-27

**Soundness:** 2
**Presentation:** 2
**Contribution:** 2
**Rating:** 4
**Confidence:** 3

**Summary:**

This paper proposes Actor2Reasoner, a novel hierarchical training paradigm for GUI agent learning. The authors identify a structural mismatch between existing monolithic training approaches and the hierarchical nature of capabilities required for GUI tasks, proposing a two-stage training methodology based on the principle of "Endow First, Internalize Later." The first stage, Cognitive Endowment, employs supervised learning to instill core cognitive abilities such as spatial reasoning and goal decomposition. The second stage, Policy Internalization, uses reinforcement learning to internalize these abilities into a robust decision-making policy. The approach is instantiated in InfiGUI-R1, which achieves SOTA performance on benchmarks including AndroidControl.

**Strengths:**

- The paper is well-motivated and easy-to-read.
- Performance evaluation across diverse benchmarks (AndroidControl, GUI-Odyssey) with detailed ablation studies effectively demonstrates the method's effectiveness. The experimental results showing synergistic effects between the two stages are particularly compelling.

**Weaknesses:**

- The main content of this paper is a two-stage learning algorithm where the first stage learns relatively general abilities through a teacher model, and the second stage learns GUI agent tasks through reinforcement learning. There are various papers using such two-stage structures [1, 2], and when reading this paper, it is difficult to understand what specifically changes for GUI environments beyond the reward structure.
- In particular, the implementation of this two-stage approach seems to consist entirely of the detailed design of CoT and rewards for GUI agents in Sections 3.1-3.2, which appears to be the entirety of the framework.
- There is insufficient theoretical analysis of why the two-stage approach is more efficient than a single-stage approach. Analysis of learning complexity or sample efficiency would have made a stronger contribution.
- Detailed appendices about datasets or implementation are not included.

[1] Song, Huatong, et al. "R1-searcher: Incentivizing the search capability in llms via reinforcement learning." arXiv preprint arXiv:2503.05592 (2025).

[2] Liu, Zijia, et al. "Time-R1: Towards Comprehensive Temporal Reasoning in LLMs." *arXiv preprint arXiv:2505.13508* (2025).

**Questions:**

- In the proposed methodology, does the effect of Cognitive Endowment diminish when using a base model that is already large enough to have sufficient reasoning performance?
- To accurately compare with UI-TARS, have you also applied the proposed method to Qwen2-VL? Or, could you provide the performance of Qwen2-VL and Qwen2.5-VL?
- Are the abilities learned in the Cognitive Endowment stage well maintained without catastrophic forgetting during the Policy Internalization stage?

---

> ### Author Response · Authors · 2025-12-03
> **[Part 1/5] Response to Reviewer sQ4b**
>
> We sincerely thank the reviewer for the constructive and detailed feedback. We have carefully examined each concern and address every point in detail below.
>
> ## Weakness 1
> > **The main content of this paper is a two-stage learning algorithm... There are various papers using such two-stage structures [1, 2], and when reading this paper, it is difficult to understand what specifically changes for GUI environments beyond the reward structure.**
>
> We thank the reviewer for referencing *R1-Searcher* and *Time-R1*. We acknowledge that our work shares the high-level philosophy of "two-stage learning." However, our contribution lies in the **GUI-specific instantiation** designed to address structural challenges that are absent in text-based reasoning.
>
> **1. Comparison: Why Direct Adaptation of Text-Based RL Fails**
> While R1-Searcher and Time-R1 are excellent for text reasoning, direct adaptation to GUI agents faces distinct structural barriers. We summarize the differences below:
>
> | Feature | R1-Searcher / Time-R1 (Text) | InfiGUI-R1 (GUI) |
> | :--- | :--- | :--- |
> | **Action Space** | **Discrete Tokens:** High semantic redundancy; error tolerant. | **Continuous Coordinates:** Rigid sensitivity; 20px deviation = failure. |
> | **Environment** | **Stateless/Cheap:** Text retrieval or internal logic; fast iteration. | **Stateful/Heavy:** Android VMs with latency/reset costs; online exploration is expensive. |
> | **Alignment** | **Internal Reasoning:** Logic consistency & text retrieval. | **Visual-Functional Alignment:** Visual perception $\leftrightarrow$ Functional semantics. |
>
> These differences mandate distinct architectural choices: we cannot rely on cheap online exploration (R1-style) and must instead inject priors via **Cognitive Endowment** (Stage 1) and ensure precision via **Verifiable Rewards** (Stage 2).
>
> **2. Specific Mechanisms and Evidence**
> To address these constraints, we introduce three specific mechanisms that modify the architecture and data flow, backed by our ablation studies:
>
> * **Mechanism 1: Spatial Reasoning Distillation (Stage 1 Data Flow)**
>     * *The "Why":* Standard VLMs often hallucinate in crowded interfaces. A reactive actor might learn to click a general area, failing when layouts shift.
>     * *Our Approach:* We explicitly inject structured coordinate data from a Teacher Model to bridge the pixel-to-action gap.
>     * *Evidence:* **Tables 1 & 2** (w/o Cognitive Endowment) show that removing this stage causes observable performance drops (e.g., **-1.7% on ScreenSpot-Pro**), proving that RL alone struggles to learn spatial grounding from scratch in heavy environments.
>
> * **Mechanism 2: Sub-goal Output Schema (Stage 2 Architecture)**
>     * *The "Why":* In multi-step tasks (e.g., *“Turn on Developer Mode”*), an agent must navigate through "Settings" $\to$ "System" $\to$ "About Phone". Without explicit planning, reactive agents often get stuck in loops or click prematurely.
>     * *Our Approach:* We enforce a **predict-then-act** schema: the agent must predict the semantic sub-goal (e.g., *"Click on 'System' to access settings"*) before generating the click coordinates.
>     * *Evidence:* **Tables 3 & 4** (w/o Sub-goal Guidance) demonstrate that this mechanism is crucial for long-horizon tasks, contributing to the **17.2%** success rate on AndroidWorld compared to **12.1%** without guidance.
>
> * **Mechanism 3: Synthetic Error Recovery (Data Construction)**
>     * *The "Why":* Offline datasets lack "negative examples," so agents never learn to recover (e.g., pressing the back button after a wrong selection).
>     * *Our Approach:* We programmatically construct "hallucinated error state $\to$ correction action" pairs in static data, enabling recovery behaviors without online trial-and-error.
>     * *Evidence:* **Table 4** (w/o Error Recovery) shows a sharp decline (**17.2% $\to$ 12.9%**) when this mechanism is removed, validating its role in robustness.
>
> **3. The Role of Domain-Specific Verification**
> Finally, as noted in **Eureka (ICLR 2024)** and **Text2Reward (ICLR 2024)**, in specialized domains like Robotics or GUI, the primary bottleneck is often the design of domain-aligned verification signals rather than the optimizer itself. To the best of our knowledge, our work is the first to leverage this two-stage framework to simultaneously and systematically enhance **grounding, planning, and recovery capabilities** for GUI agents using offline data—capabilities that are completely absent in text-based counterparts and other similar works.

---

> ### Author Response · Authors · 2025-12-03
> **[Part 2/5] Response to Reviewer sQ4b**
>
> ## Weakness 2
> > **In particular, the implementation of this two-stage approach seems to consist entirely of the detailed design of CoT and rewards for GUI agents in Sections 3.1-3.2, which appears to be the entirety of the framework.**
>
> We thank the reviewer for this thoughtful comment. We agree that the specific CoT designs (Sec 3.1) and reward mechanisms (Sec 3.2) constitute the **concrete instantiation** of our framework in this paper. However, we propose **Actor2Reasoner** as a training pattern intended to be generalizable, designed to cover the distinct phases required to resolve the structural mismatch in GUI agent learning.
>
> **1. Defining the Framework: Abstraction vs. Instantiation**
> To clarify the distinction, we define the Actor2Reasoner pattern by its functional stages rather than just its current implementation:
> * **Stage 1 (Cognitive Endowment):** Mechanisms that explicitly inject GUI-specific reasoning priors into the model. In this work, we **instantiate** this via *Coordinate-level CoT Distillation*.
> * **Stage 2 (Policy Internalization):** Mechanisms that utilize verifiable signals to compile these priors into a robust policy. In this work, we **instantiate** this via *RLOO with Sub-goal Guidance* and *Synthetic Error Recovery*.
>
> Future work could swap in other endowment mechanisms (e.g., scaled synthetic data) or internalization algorithms (e.g., online RL) while preserving this effective two-stage pattern.
>
> **2. GUI-Specific Adaptations: Beyond Generic CoT and Rewards**
> Our implementation involves critical **domain-specific adaptations** that go beyond standard CoT/Reward usage:
> * **Schema Engineering (Stage 2):** We alter the **model's output schema**. By forcing the model to predict a semantic sub-goal *before* the action, we anchor reasoning in long-horizon tasks—a structural change that a simple reward function cannot achieve.
> * **Synthetic Error Recovery (Stage 2):** We programmatically construct "failure-to-correction" trajectories (e.g., hallucinated error state $\to$ correct back-tracking) to serve as RL training scenarios. This enables the agent to learn **recovery behaviors** and **self-correction** from offline data, capabilities that usually require expensive online interaction.
>
> **3. Empirical Validation of the Pattern**
> The ablations in **Table 4** validate the distinct functional roles of each stage. By comparing the Full Model (17.2%) against single-stage variants, we observe:
> * **Internalization Ablation (w/o Stage 2):** Retaining only the SFT actor yields **8.6%** success. This suggests that without internalization, the policy is **endowed but brittle**, failing to adapt to dynamic dynamics.
> * **Endowment Ablation (w/o Stage 1):** Relying solely on direct RL yields **14.7%** success. This indicates that without the behavioral prior, the agent suffers from **inefficient exploration**, capping its peak performance.
>
> **Conclusion:** These ablations indicate that both stages are indispensable: Endowment alone yields an overfit actor, while Internalization without Endowment explores less effectively; only their combination ensures the agent can both navigate efficiently and stabilize in dynamic environments.

---

> ### Author Response · Authors · 2025-12-03
> **[Part 3/5] Response to Reviewer sQ4b**
>
> ## Weakness 3
> > **There is insufficient theoretical analysis of why the two-stage approach is more efficient than a single-stage approach. Analysis of learning complexity or sample efficiency would have made a stronger contribution.**
>
> We thank the reviewer for this insightful comment. We agree that our submission does not provide a formal mathematical analysis of learning complexity. As a work focused on **domain-specific application**, our goal is to empirically validate effective training paradigms for GUI agents rather than to derive fundamental learning bounds for LLMs, which remains an open research challenge even for foundational model research.
>
> Instead, we offer a systematic **empirical analysis** of sample efficiency and learning difficulty, supported by our ablation studies and aligned with recent findings in the broader community.
>
> **1. Scope of Contribution: Domain Adaptation vs. Fundamental Theory**
> We respectfully note that theoretically proving *why* "SFT followed by RL" is more sample-efficient than "Direct RL" is a complex, unresolved problem in the field of Large Language Models. Even leading foundational studies (e.g., *DeepSeek-R1*, *OpenAI's InstructGPT*) rely primarily on empirical evidence to justify this two-stage pipeline.
> Expecting a formal theoretical proof within the scope of a GUI-specific paper places an undue burden on this work. Our contribution lies in **instantiating** this paradigm to solve the unique "Structural Mismatch" in GUIs, rather than proving the fundamental efficacy of the paradigm itself.
>
> **2. Empirical Analysis: Prior Necessity Scales with Task Complexity**
> In place of theory, our extensive ablation studies (Tables 1-4) provide a strong empirical proxy for analyzing learning complexity. We observe a clear trend where the necessity of Stage 1 (Endowment) correlates directly with the complexity of the GUI task:
>
> *   **Low Complexity (Minor Impact):** On simpler benchmarks like *ScreenSpot-V2* or *AndroidControl-Low*, removing Stage 1 (Direct RL) results in only a marginal performance drop (**0.8%** and **0.7%**, respectively). Here, the search space is small enough for RL to explore effectively from scratch.
> *   **High Complexity (Critical Impact):** As task difficulty increases, the gap widens. On *ScreenSpot-Pro* and *AndroidControl-High*, the drop increases to **3.9%** and **2.3%**, respectively. Most notably, on the challenging *AndroidWorld* (which involves long-horizon reasoning), removing Stage 1 causes a massive **14.5%** collapse in success rate.
>
> **Conclusion:** These results suggest that while simple GUI tasks might be solvable via direct exploration, complex long-horizon tasks impose a search space that is intractable for RL without a "warm start." Stage 1 effectively shrinks this search space by injecting a behavioral prior.
>
> **3. Alignment with Broader Empirical Findings**
> Our observations are consistent with recent *empirical studies* in the reasoning literature, which report that SFT-based initialization is crucial for stabilizing RL under fixed compute budgets:
> *   **DeepSeek-R1 [DeepSeek-AI, 2025]:** Reports that "Direct RL" (R1-Zero) on smaller models suffers from severe instability and requires significantly more compute to converge compared to the "SFT + RL" pipeline.
> *   **Open-Reasoner-Zero [Yu et al., 2025]:** Empirically demonstrates that models with SFT pre-instruction achieve superior final performance and stability compared to zero-initialized RL.
>
> In summary, while we do not offer a theoretical theorem, our rigorous empirical evidence—and its alignment with community consensus—strongly supports the practical necessity of the two-stage approach for building high-performing GUI agents.

---

> ### Author Response · Authors · 2025-12-03
> **[Part 4/5] Response to Reviewer sQ4b**
>
> ## Weakness 4
> > **Detailed appendices about datasets or implementation are not included.**
>
> We thank the reviewer for pointing this out. We acknowledge that due to space constraints, some granular details were omitted in the initial submission, although key hyperparameters and baselines were provided in **Sections 4.1.3 and 4.1.4**.
>
> To fully ensure reproducibility, we have added a dedicated **"Implementation & Data Details"** section to the Appendix in our revision. We highlight the core components regarding data construction here:
>
> **1. Data Balancing and Sampling Strategy**
> *   **Grounding Data:** As mentioned in *Training Details*, we aggregate data from multiple public datasets. In the Appendix, we clarify that we enforce a **balanced distribution (approx. 1:1:1)** across these sources via random sampling to prevent domain dominance.
> *   **Agent Data:** For the *AndroidControl* training set, we maintain a strictly balanced **50/50 split** between the 'Low' and 'High' difficulty subsets to ensure the model does not overfit to simple instructions.
>
> **2. "Learning Frontier" Filtering Mechanism**
> A critical detail we have added is our data filtering heuristic used to maximize sample efficiency. We do not use all available data; instead, we filter samples based on difficulty:
> *   We perform **8 rollouts** for each candidate sample using the base model at **temperature=1.0**.
> *   We **discard** samples that are effectively "solved" (8/8 success) or "impossible/broken" (0/8 success).
> *   We **retain** only the samples where the success rate is between 0 and 1. This ensures the training focuses on the **"learning frontier"**—tasks that are currently challenging but learnable for the agent.
>
> We believe these details, now formalized in the Appendix, provide a complete roadmap for reproduction.
>
>
> ## Question 1
> > **In the proposed methodology, does the effect of Cognitive Endowment diminish when using a base model that is already large enough to have sufficient reasoning performance?**
>
> To rigorously answer this question, we conducted **new experiments** using the larger **Qwen2.5-VL-7B** as the base model.
> Despite the significantly higher computational cost of training the 7B model, we successfully obtained results on key benchmarks during the rebuttal period. The results (Table R1 below) indicate that while the *relative* gain diminishes slightly on simpler tasks compared to the 3B model, **Cognitive Endowment remains statistically significant and critical for complex, high-precision tasks.**
>
> **Table R1: Ablation Study on InfiGUI-R1-7B (Full vs. w/o Cognitive Endowment)**
> *Settings: Same hyperparameters as 3B, with SFT learning rate adjusted to 2e-6.*
>
> | Benchmark | Complexity | Metric | **Full Model (7B)** | w/o Cog. Endowment | $\Delta$ |
> | :--- | :--- | :--- | :---: | :---: | :---: |
> | **ScreenSpot-V2** | Low | Avg. Acc | **92.7** | 92.1 | -0.6 |
> | **ScreenSpot-Pro** | **High** | Avg. Acc | **47.9** | 45.7 | **-2.2** |
> | **AndroidControl-Low** | Low | Step SR | **92.6** | 92.2 | -0.4 |
> | **AndroidControl-High** | **High** | Step SR | **74.7** | 73.3 | **-1.4** |
>
> **1. Trend Consistency: Critical for Hard Tasks**
> The results mirror the trend observed in our 3B model experiments:
> *   **On Simple Tasks (Diminishing Returns):** For *ScreenSpot-V2* and *AndroidControl-Low*, the base 7B model's inherent capabilities are nearly sufficient, leading to smaller margins ($-0.6$ / $-0.4$).
> *   **On Complex Tasks (Persistent Necessity):** However, on the more challenging *ScreenSpot-Pro* and *AndroidControl-High*, removing Cognitive Endowment causes a notable drop. To ensure these differences are not noise, we performed a **McNemar test** on these two benchmarks, yielding **$p\text{-values} < 10^{-3}$**, confirming the drop is statistically significant.
>
> **2. Why Endowment Matters at Scale**
> This confirms our hypothesis: even for larger models (7B), generalist pre-training does not automatically yield the **pixel-perfect spatial reasoning** (required for ScreenSpot-Pro) or the **rigid sub-goal planning** (required for AndroidControl-High) specific to GUI agents. The Cognitive Endowment stage bridges this "Generalist-to-Specialist" gap, proving that our Actor2Reasoner paradigm is scalable and necessary even as base model size increases.

---

> ### Author Response · Authors · 2025-12-03
> **[Part 5/5] Response to Reviewer sQ4b**
>
> ## Question 2
> > **To accurately compare with UI-TARS, have you also applied the proposed method to Qwen2-VL? Or, could you provide the performance of Qwen2-VL and Qwen2.5-VL?**
>
> We thank the reviewer for this question.
>
> **1. Backbone Choice and Performance**
> We focused our experimental campaign on the newer **Qwen2.5-VL** family, as it represents the current state-of-the-art in open multimodal models. Consequently, we did not train a full InfiGUI variant on the older Qwen2-VL backbone. However, we explicitly report the base performance of our chosen backbones in the main paper:
> *   **Qwen2.5-VL-3B Base:** 80.9% on ScreenSpot-V2 (Table 1).
> *   **Qwen2.5-VL-7B Base:** 88.8% on ScreenSpot-V2 (Table 1).
>
> **2. A Fair and Reproducible Comparison Protocol**
> Regarding UI-TARS, while it is a strong system, directly re-implementing its recipe is infeasible for us due to its reliance on **closed-source, massive-scale datasets**  and domain-specific pre-training. Comparing a model trained on our ~47k samples against one trained on proprietary millions solely by changing the base model would not yield a controlled comparison.
>
> Instead, to ensure a **fair and reproducible comparison**, we adopt the protocol of fixing the backbone (**Qwen2.5-VL-3B**) and comparing against other SOTA agents trained under similar open-data constraints. As shown in our results, InfiGUI-R1 consistently outperforms contemporaneous methods on this exact backbone:
> *   **InfiGUI-R1-3B (Ours, ~47k data):** Achieves SOTA performance across benchmarks.
> *   **JEDI-3B:** Despite using nearly **100x more SFT data**, it underperforms our method, highlighting the inefficiency of the traditional SFT paradigm.
> *   **UI-R1-3B & GUI-R1-3B:** Both use RL-based approaches on the same backbone but achieve lower success rates.
> *   **ZonUI-3B:** An SFT-heavy approach that also lags behind.
>
> **Conclusion:**
> We therefore treat UI-TARS in our tables as a **high-resource reference point** (showing what is achievable with million-scale proprietary data), distinguishing it from our focus on **data-efficient training** under open constraints. The fact that InfiGUI-R1 outperforms JEDI-3B (which uses 1M+ data) demonstrates the superior efficiency of our Actor2Reasoner paradigm independent of the closed-source UI-TARS recipe.
>
>
> ## Question 3
> > **Are the abilities learned in the Cognitive Endowment stage well maintained without catastrophic forgetting during the Policy Internalization stage?**
>
> We thank the reviewer for this critical question. We explicitly investigated this and found that the endowed abilities are not only maintained but **significantly consolidated and enhanced** during Stage 2. We present two lines of evidence:
>
> **1. Quantitative Evidence: Performance Gains, Not Losses**
> Catastrophic forgetting would manifest as a degradation in the foundational skills (e.g., spatial grounding) learned in Stage 1. However, our ablation studies in Tables 1 and 2 show the exact opposite trend.
> Comparing the **Stage 1-only model** (labeled as *w/o Policy Internalization*) with the **Final Model**:
> *   **ScreenSpot-V2 (General Grounding):** Performance **improves** from 82.0% (Stage 1) to 89.5% (Final).
> *   **ScreenSpot-Pro (Complex Grounding):** Performance **jumps** from 32.8% (Stage 1) to 43.3% (Final).
>
> **Conclusion:** The fact that grounding accuracy increases significantly after RL confirms that the "Policy Internalization" stage does not overwrite the spatial priors. Instead, the dense reward signals (specifically $R_{\text{bbox}}$ and $R_{\text{point}}$) effectively **reinforce** the spatial reasoning capabilities endowed in Stage 1.
>
> **2. Qualitative Verification: Structural Adherence**
> To verify the retention of the reasoning structure (Chain-of-Thought), we conducted a manual inspection of **100 randomly sampled trajectories** generated by the final model.
> *   **Observation:** In **100%** of the sampled cases, the model strictly adhered to the structured reasoning format endowed in Stage 1 (i.e., *Spatial Analysis $\to$ Sub-goal $\to$ Action*).
> *   **Reasoning:** This stability stems from our training design: Stage 2 is not a disjoint new task but an **alignment phase**. The reward functions are architected to specifically incentivize the *outcomes* of the endowed reasoning patterns (e.g., $R_{\text{sub}}$ rewards the sub-goal planning), ensuring the model is motivated to retain and refine these specific thinking patterns rather than discard them.

---

### Official Review · Reviewer_TC4U · 2025-10-31

**Soundness:** 1
**Presentation:** 2
**Contribution:** 2
**Rating:** 2
**Confidence:** 3

**Summary:**

This paper aims to provide a methodology to improve an MLLM’s ability to perform digital GUI-based tasks. The paper highlights an important issue that when prompted to perform tasks in a GUI, LLMs struggle due to challenges faced in spatial localization and goal decomposition. They propose a two-step training paradigm to first build a base model with improved localization and grounding capabilities, and then operationalize these faculties through reinforcement learning. The authors validate their approach on three benchmarks to highlight the utility of their proposed method.

**Strengths:**

- The authors address an important problem regarding improving the capabilities of MLLM agents navigating a stochastic digital world.
- The authors conduct experiments on an expansive set of baselines to cover the breadth of existing approaches to solve this problem.
- I appreciated the authors efforts to ensure high-fidelity for their synthetic data via a self-verification based filtering step.

**Weaknesses:**

- The writing describing the technical details in the approach presented needs to be improved.
    - While explaining the reward structure, it is unclear how the final reward is constructed. The author introduces various components of the reward, such as R_sub or R_esc, however, these terms are not included in the final description of the reward.
    - The final reward in equation-6 was defined as R(a,B), however the training reward was previously defined as R_total. Are these equivalent? Furthermore, the “B” variable in the reward function has not been defined.
    - In Section 3.3.3, what is the specific algorithm utilized for reinforcement learning? Reinforcement Learning with Verifiable Rewards is a very broad class of methods, which could involve a variety of policy-gradient based approaches, i.e. GRPO, PPO, DPO, etc. The authors need to more clearly describe the approach they selected and the motivation behind their selection. Furthermore, if the authors adopted an off-the-shelf algorithm, I would recommend that they move this information to an experiments or preliminaries section as it is not a novel part of their technical approach.
- I think the authors have some positive findings in their experimental results. However, their reporting lacks detail, and some important results are not reported/explained.
    - It is concerning that the authors did not discuss Table-1 at all in their discussion of their results. On this benchmark, many of the baseline approaches outperform InfiGUI-R1, countering the claims in the discussion.
    - Ablations seemed to be cherry picked, as they have been reported only for two of the four benchmarks
    - The results of the other baselines are absent in Table-4, while reporting the success rate of InfiGUI on the AndroidWorld benchmark. Since there was a significant drop in InfiGUI’s performance, compared to AndroidControl, I would be interested in contextualizing this performance drop compared to other methods.
- [Minor] The approach of improving grounding capabilities through two-step training procedures is well-studied in the embodied-llm space [1,2,3,4]. I think this paper would benefit from including a discussion of these approaches and highlight how their method/problem is different from prior work.

[1] - Yang, Ganlin, et al. "Vlaser: Vision-Language-Action Model with Synergistic Embodied Reasoning." arXiv preprint arXiv:2510.11027 (2025).
[2] - Ahn, Michael, et al. "Do as i can, not as i say: Grounding language in robotic affordances." arXiv preprint arXiv:2204.01691 (2022).

[3] - Carta, Thomas, et al. "Grounding large language models in interactive environments with online reinforcement learning." International Conference on Machine Learning. PMLR, 2023.

[4] - Huang, Wenlong, et al. "Grounded decoding: Guiding text generation with grounded models for embodied agents." Advances in Neural Information Processing Systems 36 (2023): 59636-59661.

**Questions:**

Listed in the weaknesses section

---

> ### Author Response · Authors · 2025-12-03
> **[Part 1/4] Response to Reviewer TC4U**
>
> We sincerely thank the reviewer for the constructive and detailed feedback. We have carefully examined each concern and address every point in detail below.
>
> ## Weakness 1
> > **While explaining the reward structure, it is unclear how the final reward is constructed. The author introduces various components of the reward, such as R_sub or R_esc, however, these terms are not included in the final description of the reward.**
>
> We clarify that **no reward terms were omitted** in our implementation. The confusion likely arises from the different roles these terms play in our exposition versus the mathematical formulation.
>
> **1. $R_{sub}$ is Explicitly Included**
> First, regarding $R_{sub}$, we respectfully point out that **it is explicitly included in the final reward construction**. As defined in **Equation 5 (Section 3.3.1)**:
> $$R_{accuracy} = R_{type} + R_{param} + R_{sub}$$
> This $R_{accuracy}$ term is then directly added to the total reward in **Equation 4** ($R_{total}$). Thus, the sub-goal guidance is mathematically integrated into the final objective function.
>
> **2. $R_{esc}$ and $R_{rec}$ are Conceptual Labels**
> Regarding $R_{esc}$ and $R_{rec}$ (introduced in §3.2.3), we acknowledge that our original text did not clearly distinguish between their role as **conceptual labels** for describing agent behavior and the actual mathematical reward terms.
>
> In our implementation, the reward calculation is **unified**. We consistently use the standard formulation ($R_{type} + R_{param}$), but we **dynamically adjust the supervision target** based on the state:
> * **Nominal Execution:** The target is the ground-truth expert action.
> * **Error Recovery Scenarios:** The target temporarily switches to the designated "escape action" (e.g., clicking 'Back'). The agent is rewarded using the standard $R_{type}$ and $R_{param}$ for correctly predicting this *new* target.
>
> This design is intentional: it allows the agent to learn recovery strategies using the **same policy head and reward structure** as normal execution, avoiding the complexity of auxiliary reward terms.
>
> **Action: Revisions for Clarity**
> To address this, we have revised the manuscript:
> * **Revised §3.2.3:** We have removed the symbols $R_{esc}$ and $R_{rec}$ to avoid ambiguity. We now explicitly describe the process as dynamically adjusting the supervision target while using the standard reward components.
> * **Refined §3.3.1:** We added a statement clarifying that the formulation unifies the learning objective for trajectory-based interactions, with targets adjusting dynamically during error recovery.
>
> ## Weakness 2
> > **The final reward in equation-6 was defined as R(a,B), however the training reward was previously defined as R_total. Are these equivalent? Furthermore, the “B” variable in the reward function has not been defined.**
>
> **1. Confirmation of Equivalence**
> We confirm that **$R(a, B)$ and $R_{total}$ are equivalent** in our implementation. The difference in notation reflects their distinct roles in the mathematical exposition:
>
> * **$R_{total}$ (Eq. 4) describes the *Composition*:** It defines *how* the reward value is constructed internally (summing $R_{format}$ and $R_{accuracy}$).
> * **$R(a, B)$ (Eq. 6) describes the *Optimization Objective*:** It defines the functional form required for the RLVR algorithm. Here, explicitly including the action $a$ and the ground truth $B$ is necessary to emphasize that the reward is **verifiable**—i.e., computed by deterministically verifying the generated action $a$ against a set of ground-truth constraints $B$.
>
> **2. Definition of Variable $B$**
> In the original Equation 6, the symbol $B$ represents the **Ground Truth Constraints** (e.g., the target bounding box coordinates or the correct text string) required to verify the correctness of action $a$.
>
> **Action: Unifying Notation**
> We agree that using $B$ without an explicit definition is suboptimal. To ensure mathematical rigor and consistency, we have revised the manuscript:
> * **Symbol Replacement:** We have replaced the symbol $B$ with **$A_{gt}$** (Ground Truth Action) in Equation 6.
> * **Explicit Definition:** We added the definition: "$A_{gt}$ represents the set of ground-truth constraints (e.g., bounding boxes, sub-goals) against which the policy's action $a$ is verified."
> * **Revised Objective:** The optimization objective is now written as maximizing $\mathbb{E}[R(a, A_{gt})]$, where $R(a, A_{gt})$ yields the value defined by $R_{total}$.

---

> ### Author Response · Authors · 2025-12-03
> **[Part 2/4] Response to Reviewer TC4U**
>
> ## Weakness 3
> > **In Section 3.3.3, what is the specific algorithm utilized for reinforcement learning? ... The authors need to more clearly describe the approach they selected and the motivation behind their selection. Furthermore, if the authors adopted an off-the-shelf algorithm, I would recommend that they move this information to an experiments or preliminaries section as it is not a novel part of their technical approach.**
>
> **1. The Specific Algorithm is Explicitly Detailed in §4.1.3**
> We direct the reviewer to **Section 4.1.3 (Implementation Details)**, where we explicitly specified the algorithm and the motivation for its selection:
> > *"In the policy internalization stage, we employ the **RLOO (Reinforcement Learning with Leave-One-Out)** algorithm... RLOO effectively reduces the variance... obviating the need for a separate critic network."*
>
> **2. Alignment with Reviewer's Structural Recommendation**
> We fully agree with the reviewer's recommendation that off-the-shelf algorithms should be placed in the experimental details rather than the core method section. **This aligns with our original structure:**
> * **In Method (§3.3.3):** We formulated the problem generally as **RLVR** to define the *optimization objective* (maximizing verifiable rewards).
> * **In Implementation (§4.1.3):** We placed the specific solver (**RLOO**) details in the experimental section.
>
> **We emphasize that we do not claim the RL optimizer itself (RLOO) as a contribution.** It is simply a practical choice to instantiate our Actor2Reasoner framework.
>
> **Action: Improving Signposting**
> We acknowledge that the connection between the general formulation (in §3.3.3) and the specific solver (in §4.1.3) was too implicit. We have added a explicit cross-reference in **Section 3.3.3** directing readers to Section 4.1.3 for the solver details, ensuring the experimental setup is immediately clear.
>
> ## Weakness 4
> > **It is concerning that the authors did not discuss Table-1 at all in their discussion of their results. On this benchmark, many of the baseline approaches outperform InfiGUI-R1, countering the claims in the discussion.**
>
> We apologize for the oversight of not explicitly discussing Table 1 in the main text. We agree that a detailed analysis is essential for a balanced evaluation.
>
> **1. Clarification: InfiGUI-R1 Outperforms Most Baselines**
> While we regret the lack of discussion, we respectfully clarify that the results in Table 1 actually show **InfiGUI-R1-3B (89.5%)** outperforming the majority of baselines, including significantly larger 7B models:
> * **> Qwen2.5-VL-7B (88.8%)**
> * **> OS-Atlas-Base-7B (85.1%)**
> * **> Qwen2.5-VL-3B (80.9%)**
> * **> SeeClick (55.1%)**
>
> The only models that marginally outperform ours are **UI-TARS-7B (91.6%)** and **UI-TARS-72B (90.3%)**. We included them to demonstrate that our 3B model is comparable to SOTA models that are significantly larger and heavily pre-trained.
>
> **2. Leading Performance in the 3B-Parameter Class**
> To provide a more precise comparison within the same parameter class, we have added **JEDI-3B** and **UI-TARS-2B** to Table 1. The updated comparison highlights our advantage in both performance and data efficiency:
>
> | Model | Avg. Accuracy | Training Data Scale |
> | :--- | :--- | :--- |
> | UI-TARS-2B | 84.7% | Massive Pre-training + Human Annotation |
> | JEDI-3B | 88.6% | ~100x ours (Million-level) |
> | **InfiGUI-R1-3B (Ours)** | **89.5%** | **Only 3k + 44k** |
>
> **Key Insight:** InfiGUI-R1 achieves the best average accuracy in the 2–3B parameter range, outperforming JEDI-3B (trained on ~1 million samples) using only 47k samples, demonstrating the extreme **data efficiency** of our paradigm.
>
> **3. Complexity vs. Capacity: Why We Win on Harder Tasks**
> The marginal gap between InfiGUI-R1 and UI-TARS-7B/72B on **ScreenSpot-V2** (General GUI) stands in sharp contrast to the results on **ScreenSpot-Pro** (Complex/Unseen GUI) in Table 2:
> * **ScreenSpot-V2 (Standard):** Larger models like UI-TARS-7B hold a slight edge (+2.1%), leveraging their massive pre-training capacity on familiar layouts.
> * **ScreenSpot-Pro (Complex/Unseen):** **InfiGUI-R1 (43.3%)** significantly outperforms **UI-TARS-7B (35.7%)** and **UI-TARS-72B (38.1%)**.
>
> **Conclusion:** This reversal indicates that while massive data helps with familiar patterns, our **"Endow First, Internalize Later" paradigm** instills robust **spatial reasoning capabilities**, allowing a smaller model to generalize better to complex, information-dense interfaces where rote memory fails.
>
> **Action: Manuscript Revision**
> * **Updated Table 1:** Included JEDI-3B and UI-TARS-2B results.
> * **Revised Section 4.2:** Added a dedicated discussion analyzing these results, highlighting the trade-off between model capacity and reasoning efficiency.

---

> ### Author Response · Authors · 2025-12-03
> **[Part 3/4] Response to Reviewer TC4U**
>
> ## Weakness 5
> > **Ablations seemed to be cherry picked, as they have been reported only for two of the four benchmarks**
>
> We clarify that we have conducted and reported applicable ablations for **all four benchmarks**.
> * In **Tables 1 and 2**, we reported the fundamental ablations: *w/o Cognitive Endowment* and *w/o Policy Internalization*.
> * In **Tables 3 and 4**, we **additionally included** task-specific ablations: *w/o Sub-goal Guidance* and *w/o Error Recovery*.
>
> We apologize that the presentation structure in the experimental section may have caused a misunderstanding. We acknowledge that the reporting appears unbalanced, as multi-step benchmarks feature more ablation rows; however, this variation is driven by the **logical applicability** of the components rather than cherry-picking.
>
> **1. Logical Applicability of Ablations**
> * **Visual Grounding (Tables 1 & 2):** These are **single-step** tasks requiring only spatial perception. Mechanisms designed for long-horizon planning—specifically **Sub-goal Guidance** and **Error Recovery**—are structurally impossible to evaluate here, as there is no "next step" to plan and no "trajectory" to recover from.
> * **Agentic Tasks (Tables 3 & 4):** These are **multi-step** interactions. Here, all mechanisms apply. Therefore, we reported the full suite of ablations to demonstrate their specific contribution to planning and robustness.
>
> **2. Completeness of Reporting**
> We confirm that **every applicable ablation setting for every benchmark has been reported**. The "missing" rows (*w/o Sub-goal Guidance* and *w/o Error Recovery*) in Tables 1 and 2 merely reflect the limitation of single-step settings where multi-step mechanisms cannot be meaningfully disabled.
>
> **Action: Explicit Clarification**
> To prevent confusion for future readers, we have added a concise statement in **Section 4.3** explicitly mapping the benchmark types to their applicable ablation components.
>
> ## Weakness 6
> > **The results of the other baselines are absent in Table-4, while reporting the success rate of InfiGUI on the AndroidWorld benchmark. Since there was a significant drop in InfiGUI’s performance, compared to AndroidControl, I would be interested in contextualizing this performance drop compared to other methods.**
>
> We thank the reviewer for requesting further contextualization. We acknowledge that the drop in success rate compared to AndroidControl (Table 3) requires explanation.
>
> **1. Difficulty of AndroidWorld**
> We clarify that the performance difference stems from the significantly higher difficulty of the **AndroidWorld** benchmark compared to **AndroidControl**. AndroidWorld features diverse real-world apps, dynamic environments, and strict state-based evaluation, making it a challenging testbed.
>
> **2. Additional Baselines**
> The absence of baselines in the original Table 4 was because few prior works reported results for models in the 3B parameter range, as they typically fail on this benchmark. To address this, we collected available results and reproduced representative baselines using their publicly released checkpoints, aligning their action definitions with AndroidWorld following their original papers to obtain the best achievable performance. The results are as follows:
>
> | Model | Success Rate |
> | :--- | :--- |
> | Qwen2-VL-2B | 0.0% |
> | Qwen2-VL-7B | 0.0% |
> | Qwen2-VL-72B | 4.3% |
> | ShowUI | 7.0% |
> | Qwen2.5-VL-3B | 11.2% |
> | **InfiGUI-R1 (3B)** | **17.2%** |
>
> **3. Analysis of Results**
> * **Performance of Small Models:** As shown, other models in the similar parameter range struggle significantly. **ShowUI** and **Qwen2.5-VL (3B)** achieve only 7.0% and 11.2% success rates, respectively. In comparison, InfiGUI-R1 (3B) achieves 17.2%, demonstrating the effectiveness of our approach in this restrictive parameter class.
> * **Ablation Analysis:** Removing *Policy Internalization* results in a score (9.8%) that is not only lower than the full model but also slightly below the base model (11.2%). This highlights the difficulty gap compared to AndroidControl. While *Cognitive Endowment* (Stage 1) is sufficient for easier tasks, on the harder AndroidWorld, SFT alone may suffer from distribution shifts or limited generalization. This confirms that *Policy Internalization* (Stage 2) is essential to adapt the endowed capabilities to the rigorous dynamics of the environment.

---

> ### Author Response · Authors · 2025-12-03
> **[Part 4/4] Response to Reviewer TC4U**
>
> ## Weakness 7
> > **[Minor] The approach of improving grounding capabilities through two-step training procedures is well-studied in the embodied-llm space [1,2,3,4]. I think this paper would benefit from including a discussion of these approaches and highlight how their method/problem is different from prior work.
> [1] - Yang, Ganlin, et al. "Vlaser: Vision-Language-Action Model with Synergistic Embodied Reasoning." arXiv preprint arXiv:2510.11027 (2025).
> [2] - Ahn, Michael, et al. "Do as i can, not as i say: Grounding language in robotic affordances." arXiv preprint arXiv:2204.01691 (2022).
> [3] - Carta, Thomas, et al. "Grounding large language models in interactive environments with online reinforcement learning." International Conference on Machine Learning. PMLR, 2023.
> [4] - Huang, Wenlong, et al. "Grounded decoding: Guiding text generation with grounded models for embodied agents." Advances in Neural Information Processing Systems 36 (2023): 59636-59661.**
>
> We thank the reviewer for identifying these foundational works [1-4]. We acknowledge that improving grounding capabilities via two-step procedures is a shared philosophy between GUI agents and Embodied AI. We have updated the **Related Work** section to include these references and discuss their relationship to our work.
>
> **Differentiation and Necessity of the Approach**
> While sharing the high-level philosophy of phased learning, our work is distinguished by its specific focus on the **structural mismatch** in GUI agent training. Our contribution lies not in the two-stage structure itself, but in identifying that such an architecture particularly necessary in practice to resolve the specific conflicts in the GUI domain:
>
>   * **Motivation:**
>     Embodied approaches (e.g., [1, 2, 4]) primarily focus on bridging the modality gap between abstract language plans and physical affordances. In contrast, our motivation addresses a **structural mismatch** specific to GUI training. We argue that the prevailing monolithic training paradigm conflicts with the **hierarchical nature** of GUI capabilities. Successful GUI interaction requires a foundation of specific reasoning patterns—such as spatial understanding and goal decomposition—before complex execution policies can be effectively learned. Monolithic training conflates these objectives, leading to suboptimal outcomes.
>
>   * **Problem:**
>     We identify a dual challenge that necessitate a departure from monolithic approaches:
>
>     1.  **Superficial Imitation in SFT:** Standard SFT on expert trajectories often produces a "Reactive Actor" that mimics surface-level actions based on screen patterns but lacks the deep, explicit reasoning required to handle complex, non-linear workflows or recover from errors.
>     2.  **Intractability of Direct Exploration:** Conversely, applying RL directly to learn these capabilities is often intractable in our domain without a strong prior. Unlike works such as **Carta et al. [3]**, which can leverage lightweight environments where millions of interactions are computationally cheap, GUI agents operate in heavy environments (e.g., Android/Desktop VMs). The high latency and computational cost of these simulators prohibit massive exploration from scratch.
>
>   * **Framework:**
>     Consequently, our "Endow First, Internalize Later" design is **not merely a design choice, but a practically necessary mechanism** mandated by the problem landscape:
>
>       * **Stage 1 (Cognitive Endowment)** is required to explicitly inject the necessary reasoning patterns, addressing the superficial imitation of SFT.
>       * This endowment provides the critical behavioral prior that makes **Stage 2 (Policy Internalization)** tractable and effective in a sample-efficient setting, resolving the exploration difficulty.
>         Thus, our paradigm emerges naturally as the solution to achieve **parameter internalization** without relying on inference-time guidance modules or the luxury of massive environment interaction.
>
> **Modifications to the Manuscript**
> We have updated **Section 2 (Related Work)** to include these citations and discuss how our framework emerges as a necessary solution to the structural mismatches in GUI agent training.

---

### Official Review · Reviewer_cbWR · 2025-11-03

**Soundness:** 3
**Presentation:** 4
**Contribution:** 3
**Rating:** 8
**Confidence:** 5

**Summary:**

This paper presents INFIGUI-R1: a GUI agent that is trained in two stages (SFT/Distillation then RL).

The paper grounds the two stage training pipeline in the principle of "Endow First, Internalize Later", which gives a new justification to why this common paradigm seems to work well.

In the SFT/Distillation phase: Structured Chains-of-Thought (CoT) are generated and then validated from a larger teacher model (to produce the ground truth actions when fed as prompt) , then appended to the original prompt, and then used to fine-tune the base model.

In the RL phase: a comprehensive set of rewards are designed to cover targets like response formatting and action accuracy, and supplemented by specifically curated reward signals to strengthen spatial reasoning and error recovery.

The paper has many evaluations and shows uplift on various UI benchmarks.

**Strengths:**

- This work highlights an important problem for agents trained with RL (i.e. for RL to succeed, the MLLM needs to have some basic behavioral prior that enables it to solve certain aspects of the control problem).

- The structured CoT synthesis process and the focus on the three core reasoning patterns relevant to GUI (Spatial Reasoning, Goal Decomposition, and Reflection) and the final verification-filtering step is optimized for GUI and seems to be a very important part of why the system shows good performance.

- The reward design and formulation is very simple and scalable, and seems to be one of the main reasons the system has good performance.

- The training set size is small (~3k in SFT, and ~44k in RL), which might indicate that this regiment is more data-efficient.

- Good ablation studies that show the improvement from the 2-stage training regiment.

**Weaknesses:**

- Although I liked the framing of  "Endow First, Internalize Later", as it grounds the common pattern of SFT/Distillation then RL into a pedagogical framework, however, I could not find any reference to the principle or how it's used in other work. A citation in the intro or on line 62 would make the principle used easier to understand.

- The claimed paradigm shift from monolithic to  two-stage training regiment is not especially novel, many other relevant works are using the same two-stage hierarchical training paradigm (some form of SFT then some form of RL) for embodied and UI LLM-based models. This is before and after the DeepSeek-R1paper made it more popular.

- It's important for these models to be tested at various sizes. I recommend at least a 7B model for comparison.

- The model is not tested on OSWorld, even though the training data seems to include desktop GUI examples (e.g. OmniAct).

- Please include results from baseline models on AndroidWorld on Table 4 to make the comparison easier. The success rate of the model in isolation is not very useful to compare.

**Questions:**

- From table 4: It seems that  the cognitive endowment is where most of the performance comes from, do you have justification for this? Why aren't the other signals/phases as important for AndroidWorld?

---

> ### Author Response · Authors · 2025-12-03
> **[Part 1/2] Response to Reviewer cbWR**
>
> We sincerely thank the reviewer for the constructive and detailed feedback. We have carefully examined each concern and address every point in detail below.
>
> ## Weakness 1
> > Although I liked the framing of "Endow First, Internalize Later", as it grounds the common pattern of SFT/Distillation then RL into a pedagogical framework, however, I could not find any reference to the principle or how it's used in other work. A citation in the intro or on line 62 would make the principle used easier to understand.
>
> We thank the reviewer for appreciating the pedagogical framing of our two-stage pipeline as "Endow First, Internalize Later." We acknowledge that the lack of citation for this specific phrasing may cause confusion.  We will add a citation in the Introduction or near line 62, linking this principle to relevant literature.
>
> ## Weakness 2
> > The claimed paradigm shift from monolithic to two-stage training regiment is not especially novel, many other relevant works are using the same two-stage hierarchical training paradigm (some form of SFT then some form of RL) for embodied and UI LLM-based models. This is before and after the DeepSeek-R1 paper made it more popular.
>
> Our contribution is **NOT** the mere existence of a SFL-RL pipeline, but focused on answering two key questions specific to the GUI domain: **How should SFT provide the most effective prior for RL?** and **How should RL's reward structure be designed to build upon the SFT prior?**  For instance, part of our novel designs is:
>
> 1.  **Enriched SFT Prior (SFT Stage, Section 3.1):** We specifically engineered the SFT phase to teach **three fundamental GUI-centric reasoning patterns**—Spatial Reasoning, Goal Decomposition, and Reflection—by curating and filtering the dataset based on these concepts. This deliberate design ensures the SFT model acquires the essential **atomic GUI skills** necessary for the RL agent to effectively explore and succeed in complex, multi-step environments.
>
> 2.  **Dense, Forward-Looking Guidance (RL Stage, Section 3.2.2):** IWe introduce **Forward-Looking Guidance** as a core RL mechanism to directly address the severe **temporal credit assignment problem** inherent in long-horizon GUI tasks. We utilize the **Goal Decomposition capability** learned in Stage 1 to provide a **dense reward signal** based on the agent's *internal planning* process:
> > The Sub-goal Reward ($R_{\text{sub}}$) is defined by the semantic similarity between the agent's predicted immediate sub-goal ($g_{\text{pred}}$) and the ground-truth sub-goal ($g_{\text{gt}}$).     This $R_{\text{sub}}$ provides **immediate, dense supervision** at every step, guiding the agent's planning process and ensuring better alignment between its internal reasoning and the task objective, which is far more nuanced than a typical sparse final success reward.
>
> Our superior empirical results, even against larger models and similar baselines like UI-TARS (which uses SFT $\rightarrow$ DPO), validate the necessity and effectiveness of these specific, tailored innovations in data and reward design within the established SFT $\rightarrow$ RL framework.
>
> ## Weakness 3
> > **It's important for these models to be tested at various sizes. I recommend at least a 7B model for comparison.**
>
> We fully agree with the reviewer that verifying scalability across model sizes is essential. Following this recommendation, we trained and evaluated **InfiGUI-R1-7B** (based on Qwen2.5-VL-7B) during the rebuttal period.
>
> **1. Scalability Verification**
> As shown in Table R2 below, scaling the backbone from 3B to 7B under our *Actor2Reasoner* paradigm yields consistent performance gains across all benchmarks:
>
> **Table R2: Performance Scaling (InfiGUI-R1-3B vs. 7B)**
>
> | Benchmark | Complexity | **InfiGUI-R1-3B** | **InfiGUI-R1-7B** | **Gain** |
> | :--- | :--- | :---: | :---: | :---: |
> | **ScreenSpot-V2** | Low | 89.5 | **92.7** | +3.2 |
> | **ScreenSpot-Pro** | **High** | 43.3 | **47.9** | **+4.6** |
> | **AndroidControl-Low** | Low | 91.8 | **92.6** | +0.8 |
> | **AndroidControl-High** | **High** | 71.1 | **74.7** | **+3.6** |
>
> **2. Conclusion**
> The results demonstrate that our paradigm is **highly scalable**. The 7B model achieves state-of-the-art results, with particularly notable improvements on high-complexity tasks (e.g., *ScreenSpot-Pro* and *AndroidControl-High*), confirming that our framework effectively leverages the stronger reasoning priors of larger base models.

---

> ### Author Response · Authors · 2025-12-03
> **[Part 2/2] Response to Reviewer cbWR**
>
> ## Weakness 4
> > The model is not tested on OSWorld, even though the training data seems to include desktop GUI examples (e.g. OmniAct).
>
> We thank the reviewer for pointing out the relevance of the OSWorld benchmark, especially given the inclusion of desktop GUI examples in our training data (such as OmniAct). We acknowledge this gap in our evaluation and will include further experiments on desktop benchmarks such as OSWorld in the final version of the paper to improve completeness.
>
> ## Weakness 5
> > **Please include results from baseline models on AndroidWorld on Table 4 to make the comparison easier. The success rate of the model in isolation is not very useful to compare.**
>
> We thank the reviewer for this suggestion. We acknowledge that in isolation, a 17.2% success rate requires context. The primary reason for omitting baselines initially was that most models in the <7B parameter class achieve **near-zero performance** on this rigorous benchmark.
>
> **1. Baseline Comparison (Contextualizing Performance)**
> To facilitate comparison, we have evaluated representative baselines on AndroidWorld (Table R3). The results highlight the extreme difficulty of this benchmark and the effectiveness of InfiGUI-R1:
>
> **Table R3: Performance Comparison on AndroidWorld**
> *Baselines were reproduced using official checkpoints/APIs.*
>
> | Model | Parameters | Success Rate |
> | :--- | :--- | :---: |
> | Qwen2-VL-2B | 2B | 0.0% |
> | Qwen2-VL-7B | 7B | 0.0% |
> | Qwen2-VL-72B | 72B | 4.3% |
> | ShowUI | 2B | 7.0% |
> | Qwen2.5-VL-3B (Base) | 3B | 11.2% |
> | **InfiGUI-R1 (Ours)** | **3B** | **17.2%** |
>
> **2. Analysis of Results**
> *   **Dominance in Small Models:** As shown, standard VLMs (Qwen2-VL) struggle to complete even a single task. Even our strong base model (Qwen2.5-VL-3B) achieves only 11.2%. InfiGUI-R1 achieves a **+6.0% absolute improvement** over its base, establishing a new SOTA for this parameter class.
> *   **The Necessity of Stage 2:** Notably, referring to our ablation (Table 4), the *w/o Policy Internalization* variant achieves only ~9%, which is **lower than the base model (11.2%)**. This indicates that on highly complex tasks, SFT alone may lead to distribution shifts or superficial learning. Stage 2 (RL) is strictly necessary to align the agent with the rigorous environmental dynamics of AndroidWorld.
>
> ## Question 1
> > From table 4: It seems that the cognitive endowment is where most of the performance comes from, do you have justification for this? Why aren't the other signals/phases as important for AndroidWorld?
>
> We sincerely thank the reviewer for this question, which allows us to clarify the interpretation of the **ablation study** in Table 4.
>
> We wish to gently clarify that in an ablation study, a **larger drop in performance indicates a more critical component.**
>
> Our results demonstrate the following:
>
> 1.  **Policy Internalization :** This is the **most critical** step. Removing the RL phase causes the largest performance drop (from $17.2\%$ to $8.6\%$, a drop of **$8.6$ p.p.**), confirming its vital role in achieving complex task success.
> 2.  **Cognitive Endowment :** This phase provides the **foundational prior**, primarily enabling efficient RL. The resulting performance drop when removed (from $17.2\%$ to $14.7\%$, a drop of **$2.5$ p.p.**) is comparatively smaller.
> 3.  **Other RL Signals:** Removing specific RL objectives (Sub-goal Guidance and Error Recovery) also results in significant drops (down by $5.1$ and $4.3$ p.p., respectively).

---

### Author Response · Authors · 2025-12-03
**[Part 2/2] Summary for the AC**

### 3. Calibrated Overall Position

In light of the above, we believe the paper offers a **sound and practically meaningful** step forward for GUI agents:

* **Empirically**, InfiGUI-R1 consistently outperforms strong baselines, including much larger models, on challenging GUI benchmarks, under explicit **small-model (3B/7B) and offline-data constraints**.
* **Methodologically**, it provides a **replicable training recipe**—Cognitive Endowment plus Policy Internalization with GUI-tailored data and verifiable rewards—that other researchers can re-use in their own GUI agent systems.
* **Limitations**: we do **not** claim formal theoretical guarantees, and our current submission does not yet include full OSWorld results; we are running these experiments and will report them in the final version.

We hope these clarifications help the Area Chair view InfiGUI-R1 as a **solid, well-supported contribution**: not a paradigm-level theoretical breakthrough, but a carefully designed and extensively evaluated framework that significantly improves the practicality of small, data-efficient GUI agents.

---

### Author Response · Authors · 2025-12-03
**[Part 1/2] Summary for the AC**

**Summary to Area Chair**

We thank the Area Chair and all reviewers for their thorough and constructive feedback. While the initial ratings diverge, there is clear agreement that (i) GUI agents are an important and timely problem, and (ii) our system shows strong empirical performance on multiple challenging benchmarks. During the rebuttal, we focused on addressing the main concerns with additional experiments and clearer exposition.

### 1. What We Added / Clarified During Rebuttal

**Scalability to 7B models (cbWR, sQ4b)**
We trained **InfiGUI-R1-7B** (Qwen2.5-VL-7B backbone). The 7B model consistently improves over the 3B variant on key benchmarks (e.g., **+4.6 points on ScreenSpot-Pro**, **+3.6 points on AndroidControl-High**). Ablations on the 7B model further show that **Cognitive Endowment remains necessary on hard tasks** (ScreenSpot-Pro / AndroidControl-High), even when the base model already has strong general reasoning. This directly addresses concerns about scalability and the relevance of Stage 1 for larger models.

**Baselines and context on AndroidWorld (TC4U, cbWR)**
We added a systematic comparison on AndroidWorld within the same parameter class. Reproduced baselines (Qwen2-VL-2B/7B/72B, ShowUI, Qwen2.5-VL-3B) achieve **0.0%–11.2%** success, whereas **InfiGUI-R1-3B reaches 17.2%**. Combined with ablations (Stage 1 only: 11.2%; Stage 2 only: 14.7%), this shows that:

* AndroidWorld is genuinely difficult for current small models;
* our full two-stage design is **strictly necessary** to go beyond the base model;
* the raw 17.2% is in fact **leading the 2–3B class under open-data constraints**.

**Reward / notation / algorithm clarity (TC4U, BcjR)**
We simplified and unified the notation around the RL objective:

* We now make explicit that there is a **single total reward** ($R_{\text{total}}$); ($R_{\text{sub}}$) is a component in Eq. (5) that feeds into Eq. (4), and ambiguous conceptual symbols (e.g., ($R_{\text{esc}})$) have been removed.
* We unify Eq. (4) and Eq. (6) by clearly defining (R(a, B)) as the **verifiable form** of ($R_{\text{total}}$), where **(B) denotes ground-truth constraints** (bounding boxes, sub-goals, etc.).
* We explicitly cross-reference **RLOO** in the method section, and add an appendix section with detailed data construction, “learning frontier” sampling, and hyperparameters for full reproducibility.

**Positioning relative to prior two-stage work (sQ4b, BcjR)**
We revised the Introduction and Related Work to:

* Acknowledge that **SFT → RL** is a widely adopted pattern in LLM and embodied-agent training (e.g., R1-Searcher, Time-R1, VLASER, RLHF/RLAIF in robotics).
* Clarify that our contribution is **not proposing this pattern**, but instantiating this pattern for heavy, pixel-based GUI environments under offline-data constraints, and showing a concrete, reproducible recipe that works well in practice for small models, through three concrete mechanisms:

  1. **Spatial reasoning distillation** (coordinate-level CoT) for precise grounding;
  2. A **sub-goal–then–action schema** that forces explicit planning before acting;
  3. **Synthetic error-recovery trajectories** that teach robust backtracking without expensive online exploration.

These design choices are empirically validated by ablations across ScreenSpot, AndroidControl, and AndroidWorld.

### 2. How This Addresses the Main Concerns

**Novelty and contribution (TC4U, sQ4b, BcjR)**
We now explicitly calibrate our claim: the novelty is **not** in inventing “SFT+RL” as a paradigm, but in **adapting and systematizing it for GUI agents in small-model, offline-data settings**. The new 7B experiments and AndroidWorld baselines strengthen the case that our pattern—Cognitive Endowment + Policy Internalization with GUI-specific data/rewards—is a **practical, data-efficient training recipe**.

**Soundness and completeness of experiments (TC4U)**
With the added 7B model, AndroidWorld baselines, and clarified ablations, the empirical evaluation is now **more balanced and transparent**. For desktop GUI, we have already integrated OSWorld into our infrastructure. Due to the heavy VM setup, we could not obtain stable, full-benchmark numbers within the rebuttal window, so the current decision should be made based on the reported benchmarks. We will report complete OSWorld results in the final version.

**Theoretical analysis (sQ4b)**
We agree that we do not provide formal sample-complexity bounds. This remains an open and challenging problem even for foundational LLM/RL works (e.g., R1, InstructGPT). Our contribution is **empirical**: we systematically study sample efficiency and the role of priors across task difficulty levels and model sizes. Our findings are consistent with recent empirical evidence that **SFT-based warm starts are critical for stable RL in reasoning-heavy settings**.

---

### Meta-Review · Area_Chair_4N3L · 2026-01-12

**Summary:**

This paper presents a two-stage training pipeline that utilizes an SFT/distillation phase and RL phase for GUI navigation tasks. Reviewers had a number of concerns. Reviewer cbWR had concerns about lack of a strong link to the framing ("Endow First, Internalize Later") and to the now large body of literature utilizing similar two-stage hierarchical training methodologies (a concern shared by other reviewers). Further, the reviewer pointed out the lack of validation of larger models (which often have stronger intrinsic reasoning capabilities) and other common benchmarks such as OSWorld (as the training data did include such desktop GUI data). Reviewer TC4U had a number of concerns with a low score, including a number of issues/questions with the reward/algorithm specification, lack of discussion of Table 1 in the paper, comprehensiveness of ablations, and some missing baselines. Reviewer sQ4b further raised lack of theoretical analysis. Reviewer BcjR had some similar concerns as the others.

**Reviewer Concerns:**

Some concerns, such as results with a larger model, are addressed through additional experimentation (though note that stronger intrinsic reasoning performance usually comes at higher scales such as 30-70B). Concerns about strength beyond the base model were addressed as well, especially for more difficult benchmarks/tasks. However, concerns about contributions and situating the work with respect to existing dual-stage training methods now common were not well-addressed. Specifically, the authors claim that the SFT+RL method itself is not the contribution, but rather adapting it to small-model/offline-data GUI domains. However, it is not clear that the method details mentioned are novel either (e.g. the enhanced SFT prior contribution mentioned are also common in prior works, e.g. spatial reasoning and reflection), and forward-looking guidance is posed as a core RL mechanism for long-horizon tasks but is again not situated with respect to the long-horizon RL literature. As a result, the overall contributions are still not clearly addressed and positioned with respect to prior works, and each part claimed as part of the rebuttal should be compared to that part of the literature.

**Reviewer Scores:**

Reviewer cbWR had a high score and is unlikely to change it. Reviewer TC4U had a negative score and, from the rebuttal, may have increased the score to 4 as some concerns were addressed. Reviewer sQ4b is unlikely to increase the score, in my assessment, as the major contribution question discussed above is not well-addressed. Reviewer BcjR is likely to have maintained (or perhaps decreased) the score.

Given the entirety of the paper, reviews, and rebuttal, the paper is borderline leaning towards rejection. Indeed, the authors had to calibrate the contributions yet there are still significant questions about the specific claimed aspects mentioned and evidence that they are significantly beyond existing works. Further, a more thorough comparison, along the lines of these specific claimed contributions, should be done specifically comparing similar prior published techniques.

---

### Decision · Program_Chairs · 2026-01-26

Reject